# Adapting to *Any* Bit-Width: Channel-Wise Mixed-Precision Quantization for LLMs

**Zihan Chen**                                                                    *brf3rx@virginia.edu*
*Department of Electrical and Computer Engineering*
*University of Virginia*

**Bike Xie**                                                                          *bike@kneron.us*
*Kneron Inc.*

**Jundong Li**                                                                    *jundong@virginia.edu*
*Department of Electrical and Computer Engineering*
*University of Virginia*

**Cong Shen**                                                                      *cong@virginia.edu*
*Department of Electrical and Computer Engineering*
*University of Virginia*

**Reviewed on OpenReview:** *https://openreview.net/forum?id=1t6sEhdLxf*

## Abstract

Large Language Models (LLMs) have demonstrated remarkable success across a wide range of language tasks, but their deployment on edge devices remains challenging due to the substantial memory requirements imposed by their large parameter sizes. Weight-only quantization presents a promising solution to reduce the memory footprint of LLMs. However, existing approaches primarily focus on integer-bit quantization, limiting their adaptability to fractional-bit quantization tasks and preventing the full utilization of available storage space on devices. In this paper, we introduce Channel-Wise Mixed-Precision Quantization (CMPQ), a novel mixed-precision quantization method that allocates quantization precision in a channel-wise pattern based on activation distributions. By assigning different precision levels to different weight channels, CMPQ supports arbitrary average bit-widths in the low-bit regime (e.g., between 2 and 4 bits). CMPQ employs a non-uniform quantization strategy and incorporates two outlier extraction techniques that collaboratively preserve the critical information, thereby minimizing the quantization loss. Experiments on nine different LLMs demonstrate that CMPQ not only enhances performance in integer-bit quantization tasks but also achieves significant performance gains with a modest increase in memory usage by performing in a mixed-precision way. CMPQ represents an adaptive and effective approach to LLM quantization, offering substantial benefits across diverse device capabilities.

## 1 Introduction

Large Language Models (LLMs), trained on massive text corpora and containing up to hundreds of billions of parameters, have demonstrated exceptional performance across a wide range of language tasks (Brown, 2020; Chowdhery et al., 2023; Du et al., 2022; Hoffmann et al., 2022; Thoppilan et al., 2022; Touvron et al., 2023a; Hendrycks et al., 2021). However, deploying LLMs directly on devices for inference presents a significant challenge due to their enormous parameter sizes (Frantar & Alistarh, 2023; Xia et al., 2024; Xiao et al., 2024; Li et al., 2024b). For instance, GPT-3 (Brown, 2020), with 175 billion parameters, requires 350 GB of memory in FP16 precision, which far exceeds the capacity of the NVIDIA H100 GPU with 96 GB of memory, let alone the capabilities of edge devices.

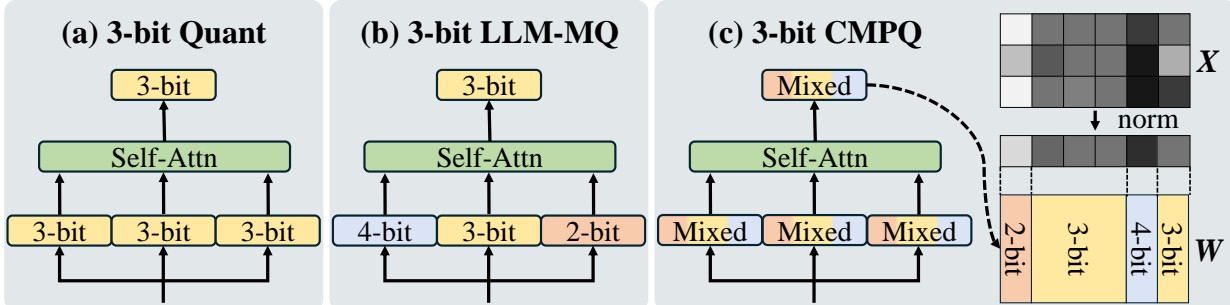

Figure 1: Illustration of different quantization approaches under a fixed bit-width constraint, such as 3 bits. (a) Standard quantization methods focus on algorithmic optimization to improve model performance, quantizing all layers uniformly to 3 bits. (b) LLM-MQ (Li et al., 2023) calculates layer-wise scores using first-order information and applies integer programming to assign lower bit-widths to less sensitive layers. (c) In contrast, our proposed CMPQ distributes the information loss evenly across layers by employing a channel-wise approach. This method assigns varying bit-widths within each layer based on activation distribution, ensuring that no single layer experiences significant information loss.

Low-precision weight-only quantization (Park et al., 2024; Lee et al., 2024; Huang et al., 2024a) has emerged as a promising solution to address this challenge by converting model weights from high bit-width representations (e.g., FP16) to lower bit-widths (e.g., 3-bit), significantly reducing the memory requirements for on-device LLM inference. In this work, we focus on Post-Training Quantization (PTQ) (Dettmers et al., 2022; Yuan et al., 2023; Shao et al., 2023; Liu et al., 2024b), which quantizes the pre-trained models without the need for retraining, a process that is often costly and resource-intensive. Most existing PTQ methods concentrate on integer-bit quantization and employ a uniform low-bit-width representation across all layers (Lin et al., 2024; Chee et al., 2024; Frantar et al., 2022), as illustrated in Figure 1(a), yielding promising performance in low-bit quantization tasks. However, these methods are limited in their adaptability to devices with additional storage capacity. For example, they are restricted to integer-bit quantization (e.g., 2-bit), even when devices could support an average precision like 2.2 bits. Attempts to utilize the additional storage, e.g., by *passively retaining more outliers in FP16*, often result in marginal performance gains with increased latency (Lin et al., 2024).

Mixed-precision quantization (Ma et al., 2023; Wang et al., 2019; Dong et al., 2019; Zhang et al., 2021) inherently supports fractional-bit quantization by allowing model weights to be quantized at different precisions. However, few works have focused on mixed-precision quantization specifically tailored for LLMs. LLM-MQ (Li et al., 2023) calculates the first-order information of layers at different bit-widths and uses integer programming to allocate layer-wise precision, as illustrated in Figure 1(b). Nevertheless, the gradient of a converged LLM is approximately zero (Kim et al., 2024), making it challenging for LLM-MQ to effectively differentiate the sensitivities of each layer. Furthermore, as demonstrated in Section 3.1, the additional quantization loss introduced by low-bit quantization can further degrade model performance.

To bridge this gap, we propose Channel-Wise Mixed-Precision Quantization (CMPQ), a PTQ framework that adapts LLMs to any average bit-width, including non-integer values, without sacrificing performance. CMPQ is based on the observation that different channels in the weight matrix have varying impacts on model performance and that assigning higher precision to salient channels can enhance the performance of quantized LLMs. Inspired by this, CMPQ performs mixed-precision quantization on a channel-wise basis, as shown in Figure 1(c). Specifically, we compute the $L_2$-norm of the activation for each layer and allocate high (or low) precision to channels with large (or small) activation norms. Further, we adopt a non-uniform quantization approach for each channel, accounting for the non-uniform nature of weight distributions. To further improve performance, we design two outlier extraction methods that separately focus on preserving activation-based outliers and quantization-aware outliers. Our contributions are summarized as follows:

- We propose CMPQ, a post-training, channel-wise mixed-precision quantization framework that supports arbitrary average bit-widths, including fractional values.

- CMPQ integrates non-uniform quantization and dual outlier handling (activation- and quantization-aware), offering fine-grained control with low overhead.

- Extensive experiments demonstrate that CMPQ (1) outperforms prior methods in both integer and fractional-bit settings, and (2) achieves substantial performance gains with modest increases in storage overhead.

## 2 Related Works

**Post-Training Quantization.** Quantization is a model compression technique that modifies the vector or matrix representations of a pre-trained model to improve inference efficiency. It can be broadly categorized into two workflows: Quantization-Aware Training (QAT) (Liu et al., 2023; Xu et al., 2024; Li et al., 2024c; Dettmers et al., 2024) and Post-Training Quantization (PTQ) (Yao et al., 2022; Xiao et al., 2023; Huang et al., 2024a; Hooper et al., 2024). QAT involves retraining the model during quantization, which is resource-intensive. In contrast, PTQ does not need retraining, making it a more feasible approach for resource-constrained scenarios (Zhou et al., 2024). In this work, we focus on weight-only PTQ methods (Lee et al., 2024; Park et al., 2024; Dettmers et al., 2023; Tseng et al., 2024). One of the early advancements in this domain is GPTQ (Frantar et al., 2022), which determines the optimal quantization order per row of the weight matrix based on reconstruction error relative to the Hessian matrix of unquantized weights. QuIP (Chee et al., 2024) further refines GPTQ by introducing an optimal adaptive method for a quadratic proxy objective. This method enhances quantization effectiveness by ensuring incoherence between the weight and Hessian matrices, achieved through random orthogonal matrix multiplication. AWQ (Lin et al., 2024) addresses the varying importance of weight channels for performance by employing a reparameterization technique, selecting coefficients via grid search to minimize reconstruction errors efficiently. SqueezeLLM (Kim et al., 2024) approaches quantization by storing outliers in a full-precision sparse matrix while applying non-uniform quantization to the remaining weights. Though promising for low-bit quantization, these methods are limited in their adaptability to devices with additional storage capacity. Specifically, they **_cannot actively_** adjust their strategy to fully utilize the available budget in fractional-bit scenarios.

**Mixed-Precision Quantization.** Uniform low-precision quantization often leads to significant accuracy drops (Habi et al., 2020; Qu et al., 2020; Hu et al., 2021). Mixed-Precision Quantization (MPQ) addresses this by assigning different bit widths to weights to improve performance (Wang et al., 2019; Li et al., 2024a; Rakka et al., 2022). While prior work focuses on small DNNs using techniques like second-order sensitivity for layer-wise MPQ (Wang et al., 2019; Howard, 2017; He et al., 2016), such methods are hard to scale to LLMs due to their high computational cost and large search space (Gholami et al., 2022). While LLM quantization methods like OWQ (Lee et al., 2024) and SqueezeLLM (Kim et al., 2024) preserve outliers in FP16 and quantize the rest with fixed integer bits, resembling mixed-precision, we classify them as integer-bit quantization since the majority of weights use fixed precision. Moreover, these strategy passively trades storage for marginal gains by storing more FP16 outliers, but this often leads to increased inference latency (Lin et al., 2024). Few studies have applied mixed-precision strategies to the majority of weights. LLM-MQ (Li et al., 2023) uses integer programming to allocate layer-wise precision based on first-order information. However, as discussed in Section 3.1, it struggles to capture the varying sensitivities of individual layers. SliM-LLM (Huang et al., 2024b) uses group-wise quantization for better performance. Their objective still focuses on average integer-bit quantization. We focus on a more challenging and practical objective: providing a highly adaptable, fine-grained post-training quantization solution that is both efficient and practical for real-world deployment. Our practical framework allows for **_continuous_** bit allocation across channels rather than being limited to integer-only schemes.

# 3 Method

Quantization typically involves mapping a continuous set of values $\boldsymbol{W}$ from a higher bit-width (e.g., 16-bit floating point) to a discrete set of values $Q(\boldsymbol{W})$ at a lower bit-width (e.g., 4-bit integers). The most commonly used uniform quantization (Krishnamoorthi, 2018) can be expressed as:

$$Q(\boldsymbol{W}) = \texttt{Clamp}\left(\frac{\boldsymbol{W}_{\texttt{FP16}} - \min(\boldsymbol{W}_{\texttt{FP16}})}{\Delta}, 0, 2^N - 1\right), \Delta = \frac{\max(\boldsymbol{W}_{\texttt{FP16}}) - \min(\boldsymbol{W}_{\texttt{FP16}})}{2^N - 1}$$

where $\Delta$ is the quantization step size, determined by the range of the original values $\boldsymbol{W}_{\texttt{FP16}}$ and the desired bit-width $N$. However, LLMs typically exhibit non-uniform weight distributions (Kim et al., 2024; Dettmers et al., 2024). In such cases, the presence of large magnitude values can lead to inefficient quantization, where certain bins or bit combinations are underutilized, resulting in suboptimal representation with few or no values assigned to some bins. To address this issue and better preserve the original weight information, we adopt non-uniform quantization. For a given vector $\boldsymbol{x} \in \mathbb{R}^n$, we compute a $2^N$ quantized representation $\boldsymbol{q} = \{q_1, \ldots, q_{2^N}\}$, and obtain its lower-precision approximation through the following procedure:

$$Q(\boldsymbol{x}) = (Q(x_1), Q(x_2), ..., Q(x_n)), \quad \text{where } Q(x_i) = \min_{q_j \in \boldsymbol{q}} |x_i - q_j|.$$

## 3.1 Preliminary Study of mixed-precision quantization

To explore the effect of mixed-precision quantization, we conduct preliminary experiments on two OPT models (Zhang et al., 2022) and compare the perplexity evaluation. In LLM-MQ (Li et al., 2023), each linear layer is associated with information loss scores under different precisions, and their approach models the average bit width as a constraint in an integer programming problem. We implement this strategy with the non-uniform quantization, and the performance is reported in Table 1 as **w/ IntProg**. Additionally, Lin et al. (2024) observe that retaining salient weights based on activation distributions in higher precision significantly enhances quantized performance. For each layer, we compute the channel-wise $L_2$-norm of activations and select the top (and bottom) $k\%$ of channels. These channels are quantized to 4-bit (high precision) or 2-bit (low precision), respectively. Results for $k = 1$ and $k = 10$ are presented in Table 1.

Table 1: Preliminary of various mixed-precision 3-bit quantization methods. **IntProg** denotes quantization method in LLM-MQ. **10% (1%)2b, 10% (1%)4b** indicates quantization where 10% (1%) of channels are 4-bit and 10% (1%) are 2-bit, based on the activation distribution.

| Method | OPT-2.7 | | OPT-6.7 | |
|---|---|---|---|---|
| | Wiki ($\downarrow$) | C4 ($\downarrow$) | Wiki ($\downarrow$) | C4 ($\downarrow$) |
| **3-bit** | 13.45 | 14.08 | 11.48 | 12.28 |
| **IntProg** | 14.75 | 14.88 | 12.69 | 13.13 |
| **10%2b, 10%4b** | 13.53 | 14.27 | 11.61 | 12.42 |
| **1%2b, 1%4b** | **13.38** | **14.05** | **11.45** | **12.26** |

From Table 1, we observe that applying integer programming and assigning a fixed bit-width to each layer is insufficient for achieving effective mixed-precision quantization, and may even perform worse than using 3-bit quantization across all layers. This could be attributed to two factors: (i) the per-layer scores defined in Li et al. (2023) struggle to effectively distinguish the sensitivity of different layers, as the gradients in a converged LLM are nearly zero, and (ii) the additional information loss introduced by 2-bit quantization outweighs the compensatory gains from 4-bit quantization when compared to 3-bit quantization (Chee et al., 2024). On the other hand, quantizing each layer to different precisions based on activation distributions can yield better results than consistently using 3-bit across all layers. However, extending mixed-precision quantization to more channels could degrade performance, due to the same issue outlined in (ii).

## 3.2 Channel-Wise Mixed-Precision Quantization

Our primary objective is to develop an algorithm that effectively utilizes mixed-precision quantization to adaptively compress LLMs under *any* given average bit constraint, including fractional bit-widths. Additionally, we still aim to achieve strong performance on integer-bit quantization tasks, such as 3-bit quantization, compared with existing works. To this end, we propose Channel-Wise Mixed-Precision Quantization (CMPQ). In this section, we first introduce the channel-wise non-uniform quantization method, followed by a detailed explanation of our outlier protection strategy.

### 3.2.1 Channel-Wise Non-Uniform Quantization

From the observations in Section 3.1, we can draw two key intuitions: ❶ channel-wise mixed-precision quantization can enhance model performance compared to layer-wise mixed-precision quantization, and ❷ when implementing channel-wise quantization, it is advisable to limit the number of 2-bit channels to minimize the information loss. Based on these, we propose channel-wise non-uniform quantization.

Research has shown that the weight distributions in LLM layers exhibit non-uniform patterns (Dettmers et al., 2024). Previous approaches have focused on uniform quantization (Chee et al., 2024; Frantar et al., 2022), which divides the weight range into evenly spaced bins. However, this approach is suboptimal, as it fails to account for the non-uniform weight distributions, and struggles to improve end-to-end latency in memory-bound LLM inference (Kim et al., 2024). Following Kim et al. (2024), we adopt non-uniform quantization. Specifically, for each channel $\boldsymbol{W}_{i,:}$ in the weight matrix $\boldsymbol{W} \in \mathbb{R}^{d_{in} \times d_{out}}$, we apply a $K$-means clustering algorithm, where the value of $K$ is determined by the precision assigned to the channel (e.g., $K = 8$ for 3-bit quantization). After clustering, each weight in $\boldsymbol{W}$ is represented by its nearest centroid from the set of $K$ centroids $\{q_1, \ldots, q_K\}$.

We draw inspiration from our preliminary studies to determine the precision for each channel through a simple yet effective approach. We sample a calibration set to perform forward propagation on the LLMs, obtaining the activation matrix $\boldsymbol{X} \in \mathbb{R}^{n \times d_{in}}$ for each layer weight matrix $\boldsymbol{W}$. The per-channel $L_2$-norm of $\boldsymbol{X}$ is then computed, yielding a 1-dimensional vector $\boldsymbol{a} \in \mathbb{R}^{d_{in}}$. Based on this, we calculate the channel precision allocation vector $\boldsymbol{c}$ as described in Algorithm 1, and apply non-uniform quantization to quantize each channel $\boldsymbol{W}_{i,:}$ to $c_i$ bits. The key idea is that when the average bit-width $b$ exceeds 3, we focus on quantizing the most salient channels, determined by the activation distribution, into higher precision to enhance model performance. Conversely, when the average bit-width $b$ is below 3, we concentrate on quantizing less

---

**Algorithm 1** Channel-wise Precision Allocation

**Require:** $\boldsymbol{a} \in \mathbb{R}^{d_{in}}$, average bit-width $b \in [2, 4]$
**Ensure:** $\boldsymbol{c} \in \mathbb{R}^{d_{in}}$
  $\boldsymbol{c} \leftarrow 3 \cdot \mathbf{1}$
  **if** $b > 3$ **then**
    $q \leftarrow 1 - (b - 3)$; $l \leftarrow q$-quantile of $\boldsymbol{a}$
    $\boldsymbol{c}[\boldsymbol{a} > l] \leftarrow 4$
  **else if** $b < 3$ **then**
    $q \leftarrow 3 - b$; $s \leftarrow q$-quantile of $\boldsymbol{a}$
    $\boldsymbol{c}[\boldsymbol{a} < s] \leftarrow 2$
  **else**
    $s \leftarrow$ 1st, $l \leftarrow$ 99th quantile of $\boldsymbol{a}$
    $\boldsymbol{c}[\boldsymbol{a} < s] \leftarrow 2$; $\boldsymbol{c}[\boldsymbol{a} > l] \leftarrow 4$
  **end if**

---

critical channels into lower precision to minimize quantization loss. For 3-bit quantization, we protect approximately 1% of the salient weight channels by assigning them 4-bit precision for improved performance, as motivated in Table 1.

It is worth noting that we intentionally avoid protecting additional channels in high precision and incorporating equivalent channels in low precision for compensation due to the following reasons: (i) Only a small fraction of weights (<1%) are salient (Lin et al., 2024), and protecting them would significantly reduce quantization loss. (ii) The additional information loss incurred by low precision outweighs the compensatory benefits gained from high precision.

### 3.2.2 Outlier Protection

Another key challenge in low-bit LLM quantization is the protection of outlier values (Bondarenko et al., 2021; Wei et al., 2022; Dettmers et al., 2022). Previous studies have demonstrated that naively quantizing weights with a large dynamic range significantly degrades performance, particularly at low precisions (Kim

et al., 2024). However, in some cases, retaining a small fraction (less than 1%) of outlier weights in FP16 has been shown to reduce up to 75% of the total quantization error (Dettmers et al., 2023). This suggests that extracting outliers prior to quantization can mitigate their negative impact and minimize quantization loss. Consistent with prior works (Li et al., 2023; Kim et al., 2024), we retain 0.5% of outliers in high precision (16-bit), while applying quantization to the remaining weights. Our method further distinguishes between two categories of outliers: those identified by activation magnitude and those based on quantization error.

As illustrated in Figure 2, we observe that outliers exhibit a channel-wise pattern, if an outlier appears in a channel, it consistently occurs across all tokens. Table 1 demonstrates that preserving salient weights based on the activation distribution helps mitigate quantization loss. Motivated by these findings, we introduce an activation-based outlier detection method, which identifies outliers $\boldsymbol{O}_{\mathrm{act}}$ from the weight matrix $\boldsymbol{W}$. Specifically, we select channels corresponding to the top 0.45% largest values in the activation's L2-norm vector $\boldsymbol{a}$ and preserve these channels in FP16 precision.

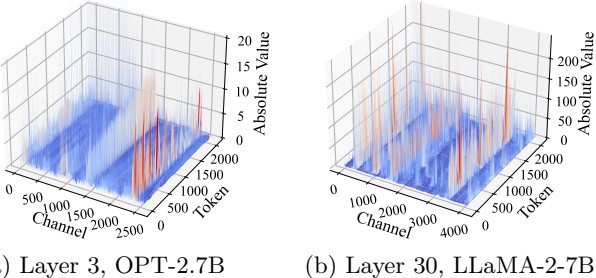

(a) Layer 3, OPT-2.7B  (b) Layer 30, LLaMA-2-7B

Figure 2: Absolute activation magnitudes of the out projection layer for different language models.

In addition to selecting channel-wise outliers, we also investigate the protection of a small subset of quantization-sensitive outliers. Though we introduced our non-uniform quantization method, a small fraction of weights exhibit significantly larger magnitudes compared to the majority. These large weights can distort the clustering process by shifting centroids away from the bulk of the weight distribution, thereby negatively impacting the performance of the quantized LLM. A conventional approach to outlier protection involves the removal of weights based solely on their magnitude. However, instead of simply eliminating high-magnitude outliers, our objective is to identify and remove those that most adversely affect the quantization process.

To achieve this, we apply another $K$-means clustering step prior to quantization. Specifically, given $\boldsymbol{W}' = \boldsymbol{W} - \boldsymbol{O}_{\mathrm{act}}$ that represents the remaining weights after activation outlier removal, we use Algorithm 1 to determine the channel precisions $\boldsymbol{c}$. We then apply channel-wise non-uniform quantization based on $\boldsymbol{c}$ to obtain the quantized model $\boldsymbol{W}'_q$. We identify the set of outliers $\boldsymbol{O}_{\mathrm{q}}$ in $\boldsymbol{W}'$ corresponding to the top 0.05% of the largest values in $|\boldsymbol{W}' - \boldsymbol{W}'_q|$. This approach preserves these magnitude-based outliers in FP16 format, not only to mitigate their influence on model output but also to ensure that the centroids $\{q_1, \ldots, q_K\}$ better represent the majority of the weights, rather than being skewed by a small number of outliers. Two types of outliers $\boldsymbol{O} = \boldsymbol{O}_{\mathrm{act}} + \boldsymbol{O}_{\mathrm{q}}$ are removed from the weight matrix $\boldsymbol{W}$, and the remaining weights then undergo the quantization process to obtain $\boldsymbol{W}_q$. $\boldsymbol{W}_q + \boldsymbol{O}$ is used as model weights for the final inference. Notably, the overhead associated with this decomposition is minimal, as the number of outlier values is relatively small, typically around 0.5% of the total values.

### 3.2.3 Discussion: Efficiency and Novelty of CMPQ

CMPQ is designed to be both practically efficient and conceptually novel, targeting real-world deployment scenarios where computational and memory budgets are constrained.

From an efficiency standpoint, CMPQ only requires forward propagation and does not depend on backpropagation, which is necessary for many existing quantization techniques (Li et al., 2023; Kim et al., 2024). Consequently, the memory requirements for our proposed CMPQ during quantization are moderate; for instance, loading the OPT-6.7B model necessitates 24.8 GB of memory, whereas the backward pass for the same model requires 49.61 GB of memory. Additionally, CMPQ has minimal reliance on the calibration set, as it only measures the $L_2$-norm per channel, thereby mitigating the risk of overfitting. For a comparison with backpropagation-dependent methods, refer to Appendix C.2, and for an analysis of CMPQ's robustness with respect to the calibration dataset, see Section 4.5.

Beyond efficiency, CMPQ's core novelty lies in its goal of delivering a highly adaptable, fine-grained post-training quantization solution suitable for real-world deployment. Unlike prior methods restricted to fixed

Table 2: Perplexity (↓) comparison of OPT and LLaMA2 models quantized to {2, 3, 4}-bit precision using various quantization methods on the C4 and WikiText-2 datasets. Bold font indicates the best performance across all methods, while underlined results denote the second-best.

| Method | Avg. Bit | OPT-2.7B | | OPT-6.7B | | LLaMA2-7B | | LLaMA2-13B | |
|---|---|---|---|---|---|---|---|---|---|
| | | Wiki | C4 | Wiki | C4 | Wiki | C4 | Wiki | C4 |
| FP16 | 16 | 12.47 | 13.17 | 10.86 | 11.74 | 5.47 | 6.97 | 4.88 | 6.47 |
| RTN | 2 | >100 | >100 | >100 | >100 | >100 | >100 | >100 | >100 |
| GPTQ | 2 | >100 | >100 | >100 | >100 | 36.77 | 35.7 | 13.67 | 16.45 |
| AWQ | 2 | – | – | – | – | >100 | >100 | >100 | >100 |
| QuIP | 2 | >100 | 38.07 | 22.33 | 21.62 | 27.12 | 31.33 | 10.09 | 13.13 |
| QuIP# | 2 | – | – | – | – | **12.30** | **14.80** | **7.60** | **9.57** |
| LLM-MQ | 2 | >100 | >100 | >100 | >100 | 96.61 | 85.16 | 15.69 | 17.02 |
| CMPQ | 2 | **32.46** | **28.32** | **18.63** | **18.31** | 14.37 | 15.97 | 9.14 | 11.25 |
| RTN | 3 | >100 | >100 | >100 | >100 | >100 | >100 | 10.68 | 12.50 |
| GPTQ | 3 | 17.09 | 18.14 | 14.87 | 17.13 | 6.25 | 7.97 | 6.17 | 7.06 |
| AWQ | 3 | 16.32 | 16.28 | **11.41** | 12.30 | 6.24 | 7.84 | **5.32** | 6.94 |
| QuIP | 3 | 17.44 | 15.63 | 11.51 | 13.30 | 6.80 | 7.75 | 5.65 | 7.25 |
| QuIP# | 3 | – | – | – | – | 6.19 | 7.85 | 5.34 | 6.98 |
| GPTVQ | 3 | – | – | – | – | 6.19 | 7.86 | 5.41 | 7.05 |
| VPTQ | 3 | – | – | – | – | 6.17 | 7.67 | 5.42 | 7.03 |
| LLM-MQ | 3 | 18.09 | 17.42 | 16.01 | 16.51 | 7.16 | 8.94 | 5.89 | 7.64 |
| CMPQ | 3 | **13.38** | **14.05** | 11.45 | **12.26** | **6.14** | **7.66** | 5.34 | **6.93** |
| RTN | 4 | 16.69 | 18.75 | 12.15 | 14.40 | 6.12 | 7.72 | 5.20 | 6.83 |
| GPTQ | 4 | 12.93 | 14.99 | 11.49 | 13.16 | 5.72 | 7.23 | 5.08 | 6.74 |
| AWQ | 4 | 12.73 | 13.48 | **10.93** | 11.86 | 5.72 | 7.13 | 4.98 | 6.56 |
| QuIP | 4 | 12.69 | 14.55 | 10.98 | 12.86 | 5.72 | 7.12 | 5.29 | 6.83 |
| QuIP# | 4 | – | – | – | – | 5.66 | 7.17 | 5.00 | 6.59 |
| GPTVQ | 4 | – | – | – | – | 5.68 | 7.25 | 5.68 | 7.26 |
| VPTQ | 4 | – | – | – | – | 5.64 | 7.13 | **4.96** | 6.59 |
| LLM-MQ | 4 | 13.06 | 13.70 | 11.04 | 12.22 | 5.68 | 7.22 | 4.98 | 6.58 |
| CMPQ | 4 | **12.63** | **13.33** | 10.95 | **11.83** | **5.61** | **7.10** | 4.98 | **6.55** |

integer-bit schemes, CMPQ supports ***continuous*** channel-wise bit allocation, enabling structured use of fractional-bit precision (e.g., 2.4 or 3.8 bits) for tighter storage control. Rather than passively increasing FP16 outlier preservation as storage grows, as in previous methods, CMPQ proactively allocates precision to low-bit channels while keeping the FP16 footprint fixed. This strategy achieves better memory-accuracy trade-offs without significantly increasing inference latency, addressing a key limitation of prior mixed-precision approaches.

# 4 Experiments

## 4.1 Experiment Setup

**LLM Models and Datasets.** We perform our main experiments on two models from the OPT family (Zhang et al., 2022) (OPT-2.7B and OPT-6.7B) and two models from the LLaMA2 family (Touvron et al., 2023b) (LLaMA2-7B and LLaMA2-13B). The evaluation of the quantized models is based on perplexity across two language generation tasks, WikiText-2 (Merity et al., 2016) and C4 (Raffel et al., 2020), as perplexity is a widely recognized metric for assessing the LLM's performance. For calibration, we follow previous works (Chee et al., 2024; Frantar et al., 2022; Kim et al., 2024) and use a set of 128 randomly selected 2048-token segments from the C4 dataset, which contains generic text data from web crawls. This

ensures consistency in comparison with baselines and avoids the use of task-specific data when quantizing other datasets. Experiments in the main results are implemented in PyTorch (Paszke et al., 2019) and executed on two A6000 GPUs, with performance monitoring handled by the Torch CUDA profiler.

We extend our evaluation to include quantization results for the other four OPT and LLaMA2 models, scaling up to 70B parameters, as well as a newer model, LLaMA3-8B (AI@Meta, 2024), with results presented in Appendix B. We further evaluate commonsense QA benchmarks PIQA (Bisk et al., 2020) and HellaSwag (Zellers et al., 2019), as well as in-context learning ability using MMLU (Hendrycks et al., 2021) in a few-shot setting.

**Baselines.** We evaluate CMPQ against a comprehensive set of PTQ baselines, including integer-bit methods such as Round-to-Nearest (RTN), GPTQ (Frantar et al., 2022), AWQ (Lin et al., 2024), QuIP (Chee et al., 2024), QuIP# (Tseng et al., 2024), as well as two recent state-of-the-art PTQ methods, GPTVQ (Van Baalen et al., 2024) and VPTQ (Liu et al., 2024a). We also compare against the mixed-precision method LLM-MQ (Li et al., 2023). For integer-bit baselines, we restrict the comparison to standard {2, 3, 4}-bit settings, where they are typically designed to operate.

To ensure fair comparison, we standardize the outlier handling strategy across all methods. Specifically, LLM-MQ retains 0.5% outliers in FP16 precision. For GPTQ, AWQ, and QuIP, we adopt a group size of 128, which has been shown to yield a similar average bit-width to retaining 0.5% FP16 outliers (Kim et al., 2024). In CMPQ, the same outlier protection ratio (0.5%) is maintained, divided between activation-based and quantization-aware outliers, enabling comparisons under equivalent storage constraints.

To broaden the experimental coverage, we further include comparisons with one recent mixed-precision method, SliM-LLM (Huang et al., 2024b) and a k-means based PTQ approach SqueezeLLM (Kim et al., 2024). These results are presented in Appendix C. and provide a comprehensive evaluation across both earlier and more recent quantization techniques for LLMs.

**Latency Profiling.** We measure the latency and peak memory usage for generating 128 tokens on an A6000 GPU using the Torch CUDA profiler. Following Kim et al. (2024), we implement a standard kernel for single-batch inference based on the widely used open-source codebase GPTQ-For-LLaMA. In general, we demonstrate that CMPQ achieves latency and memory usage comparable to other integer bit-width quantization methods like SqueezeLLM (Kim et al., 2024).

## 4.2 Main Results

Table 2 presents the main results comparing CMPQ with a broad range of post-training quantization baselines. Overall, CMPQ achieves strong and competitive performance across models and bit-width settings, and in many cases outperforms existing baselines. In particular, CMPQ shows clear advantages in several low-bit settings, especially under 3-bit quantization with C4 as the calibration set. At 2 bits, while specialized methods such as QuIP and QuIP# remain very strong baselines, CMPQ still delivers competitive results despite relying only on channel-wise activation $L_2$-norms rather than more complex sensitivity estimation. This suggests that CMPQ is relatively robust to the choice of calibration data and can generalize well across dataset distributions. In addition, although LLM-MQ is designed for mixed-precision quantization, it often struggles to achieve comparable performance, likely due to the limitations of its first-order sensitivity estimates in converged LLMs. The strong performance of CMPQ across tasks is driven by the combination of channel-wise non-uniform quantization, outlier protection, and mixed-precision allocation, with the latter providing further gains in the 3-bit setting.

## 4.3 Fractional Bit Quantization

In Figure 3, we compare the C4 perplexity of LLMs quantized by CMPQ and LLM-MQ under fractional bit-width constraints. First, it is evident that CMPQ consistently outperforms LLM-MQ across various bit-widths and models. LLM-MQ quantizes entire layers to a fixed precision, which can result in significant information loss within a single layer, negatively impacting the overall model performance. In contrast, CMPQ allocates mixed-precision in a channel-wise manner, distributing the information loss more evenly across layers and avoiding a substantial loss in any single layer. Another key observation, visible when

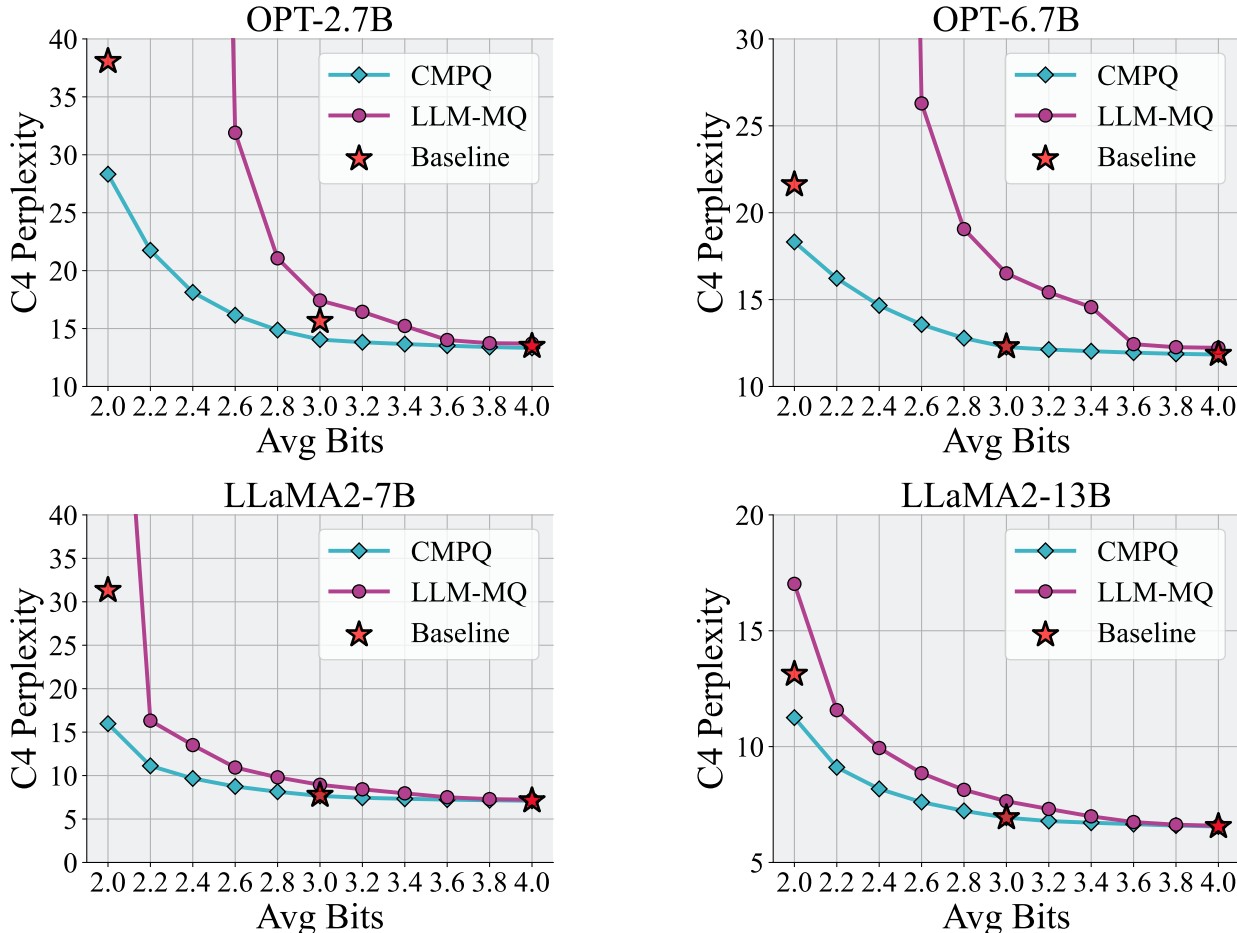

Figure 3: Comparison of C4 perplexity between CMPQ and LLM-MQ for fractional bit-width quantization. The red star indicates the best performance achieved by integer-based baselines at {2, 3, and 4} bits.

examining the transition from 2-bit to 2.2-bit quantization, highlights the advantage of mixed-precision. Introducing just a 10% increase in storage overhead at lower bit-widths can lead to significant performance gains. For instance, in the case of LLaMA2-7B, CMPQ yields a 30% improvement in perplexity (from 15.97 to 11.11), while LLM-MQ also shows a dramatic improvement, moving from poor performance (85.16) to performance that even surpasses the baseline (16.32). This demonstrates that mixed-precision quantization can trade a small increase in storage overhead for a substantial boost in performance – something that is not achievable with integer-only bit quantization methods.

As the average bit-width increases, the performance of both methods converges and approaches that of the baseline at 4-bit quantization. This indicates that quantization techniques face greater challenges at lower bit-widths. The strong performance of CMPQ at 2-bit and 3-bit quantization demonstrates its effectiveness, particularly in scenarios where lower bit-widths are required. We observe similar conclusions on WikiText-2 as well, and report the corresponding comparison in Appendix B.4.

## 4.4 Ablation Study of CMPQ Components

To assess the contribution of each component in CMPQ, we perform a step-by-step ablation starting from the RTN baseline, incrementally adding key modules under 3-bit quantization. The results are presented in Table 3. We find that outlier protection and non-uniform quantization contribute the most significant improvements in performance. Although the mixed-precision component yields a smaller direct gain, it

is essential for enabling flexible adaptation to diverse storage budgets—an important factor for practical deployment scenarios.

Table 3: Ablation results of CMPQ components on OPT-2.7B and LLaMA2-7B. The performance is evaluated on WikiText-2 dataset.

| Model | RTN | $+O_{\mathrm{act}}$ | +Non-uniform | +Mixed-Precision | $+O_q$ (CMPQ) |
|---|---|---|---|---|---|
| OPT-2.7B | >100 | 31.57 | 14.20 | 13.42 | 13.38 |
| LLaMA2-7B | >100 | 8.53 | 6.45 | 6.21 | 6.14 |

## 4.5 Robustness to the Calibration Set

We evaluate the robustness of CMPQ by analyzing its performance using different calibration sets. Specifically, we compare CMPQ with QuIP in quantizing two OPT models into 2-bit representations. The results, presented in Table 4, demonstrate that CMPQ consistently outperforms QuIP in low-bit quantization across different calibration sets.

As the size of LLMs increases, both methods demonstrate improved robustness. However, a notable difference emerges in their performance under varying conditions. While QuIP performs effectively on 2-bit quantization when the evaluation dataset aligns with the calibration set (see Table 2), it experiences a significant performance decline and may even diverge when tested on a different dataset. This decline can be attributed to QuIP's dependence on the Hessian matrix derived from the calibration set, which makes it highly sensitive to changes in the dataset. In contrast, CMPQ exhibits greater resilience, relying only on the average activation $L_2$-norm from the calibration set – a measure that generalizes more robustly across different datasets.

Table 4: Robustness analysis of the calibration dataset. We compare CMPQ with QuIP for 2-bit quantization tasks and report the perplexity differences across two different calibration datasets.

| OPT 
 Cali | Eval | QuIP | | CMPQ | |
|---|---|---|---|---|---|
| | | Wiki | C4 | Wiki | C4 |
| **2.7B** | Wiki | 32.84 | 242.69 (+204.62) | 29.62 | 27.56 (-0.76) |
| | C4 | >1000(div) | 38.07 | 32.46(+2.84) | 28.32 |
| **6.7B** | Wiki | 22.16 | 107.56 (+85.94) | 18.86 | 18.39 (+0.08) |
| | C4 | 22.33(+0.17) | 21.62 | 18.63 (-0.23) | 18.31 |

## 4.6 Ablation Study of Outlier Protection

In this section, we conduct experiments on the OPT-2.7B and OPT-6.7B models to analyze the impact of two distinct types of outliers on quantization performance. For a consistent comparison, when we remove one type of outlier, we retain the other type, ensuring that it constitutes 0.5% of the entire weight matrix. The results, presented in Table 5, demonstrate that both types of outliers contribute to enhancing the performance of quantized LLMs, particularly in low-bit settings. Notably, protecting both types of outliers yields the best results in general. This is because each type of outlier addresses different aspects: activation-based outliers safeguard salient weights, while quantization-based outliers ensure that the clustering process during quantization is not distorted by extreme values, thereby focusing on the majority of weights. In summary, these two strategies complement each other, working in tandem to improve the overall model performance.

## 4.7 Additional Experimental Results on More Downstream Tasks

To further validate the effectiveness of CMPQ on more complex tasks, we conduct additional experiments with LLaMA2-7B on benchmarks that assess broader real-world capabilities. Specifically, we evaluate zero-

Table 5: Ablation of the outlier protection strategies. Best performances are in bold, with underlined text showing the second best.

| Method | Avg. Bit | OPT-2.7B | | OPT-6.7B | |
|---|---|---|---|---|---|
| | | Wiki | C4 | Wiki | C4 |
| w/o Outlier | 3 | 13.84 | 14.47 | 11.58 | 12.49 |
| w/o $O_q$ | 3 | 13.59 | 14.28 | 11.46 | 12.36 |
| w/o $O_{act}$ | 3 | 13.39 | 14.05 | 11.49 | **12.23** |
| CMPQ | 3 | **13.38** | **14.05** | **11.45** | 12.26 |
| w/o Outlier | 4 | 12.87 | 13.44 | 11.18 | 11.94 |
| w/o $O_q$ | 4 | 12.68 | 13.38 | 10.96 | 11.85 |
| w/o $O_{act}$ | 4 | 12.69 | 13.34 | 11.06 | 11.85 |
| CMPQ | 4 | **12.63** | **13.33** | **10.95** | **11.83** |

shot performance on commonsense QA benchmarks PIQA (Bisk et al., 2020) and HellaSwag (Zellers et al., 2019), and five-shot in-context learning performance on MMLU (Hendrycks et al., 2021). The results, presented in Table 6, illustrate that CMPQ consistently achieves higher accuracy across various bit-widths compared to all baselines. This demonstrates that CMPQ not only excels in perplexity evaluation but also performs robustly in more challenging scenarios.

Table 6: Quantizing LLaMA2-7B with various post-training quantization methods, and evaluating on zero-shot commonsense QA benchmarks and five-shot MMLU benchmark

| Method | Avg. Bit | HellaSwag | PIQA | MMLU | | | | |
|---|---|---|---|---|---|---|---|---|
| | | | | STEM | Social | Humanities | Other | **Average** |
| **FP16** | 16 | 57.10 | 78.07 | 37.31 | 52.91 | 43.04 | 53.82 | 46.46 |
| **RTN** | 3 | 51.68 | 73.28 | 27.93 | 23.63 | 24.87 | 24.12 | 25.08 |
| **GPTQ** | 3 | 49.05 | 73.23 | 28.24 | 25.67 | 31.69 | 32.92 | 29.85 |
| **AWQ** | 3 | – | – | 31.74 | 36.59 | 30.86 | 38.09 | 33.98 |
| **QuIP** | 3 | 53.41 | 75.12 | 27.27 | 30.81 | 27.08 | 25.51 | 27.57 |
| **CMPQ** | 3 | **53.85** | **75.70** | 32.54 | 43.26 | 36.34 | 45.99 | **39.27** |
| **RTN** | 4 | 53.70 | 77.58 | 34.59 | 46.05 | 39.62 | 47.78 | 41.83 |
| **GPTQ** | 4 | 55.99 | 77.48 | 34.13 | 43.87 | 38.13 | 45.59 | 40.25 |
| **AWQ** | 4 | 56.40 | 77.31 | 36.24 | 50.31 | 41.72 | 52.22 | 44.85 |
| **QuIP** | 4 | 55.74 | 77.73 | 36.25 | 48.07 | 39.83 | 49.07 | 43 |
| **CMPQ** | 4 | **56.75** | **77.93** | 36.48 | 50.57 | 41.91 | 52.16 | **45.00** |

## 5 Quantization Cost Analysis

### 5.1 Memory Usage for Quantized LLMs

For outlier handling, we adopt the same approach as baseline methods like LLM-MQ (Li et al., 2023), storing 0.5% of outliers to ensure a fair comparison. Regarding the additional memory usage, for instance, in LLaMA2-7B (hidden dim = 4096), we store one integer per channel for bit allocation and eight cluster centers (look-up-table). This results in an overhead of approximately $(1 \times 4 + 8 \times 16)/4096 \approx 0.03$ bits per weight, which is mild. As the hidden dimension increases, this overhead becomes even less significant. In Table 7, we also present a CMPQ variant using 4-bit precision for cluster centers, quantizing the look-up-table (LUT) from FP16 to 4-bit. This reduces the overhead to $(1 \times 4 + 8 \times 4)/4096 < 0.01$ bits per weight, with only a minor performance drop of less than 1.5% ($(6.23 - 6.14)/6.14 \approx 1.5\%$).

Table 7: Performance of different models with varying LUT precision on WikiText-2 and C4 datasets.

| Model | LUT-bit | WikiText-2 | C4 |
|---|---|---|---|
| LLaMA2-7B | 4 | 6.23 | 7.71 |
| LLaMA2-7B | 16 | 6.14 | 7.66 |
| LLaMA2-13B | 4 | 5.39 | 6.95 |
| LLaMA2-13B | 16 | 5.34 | 6.93 |

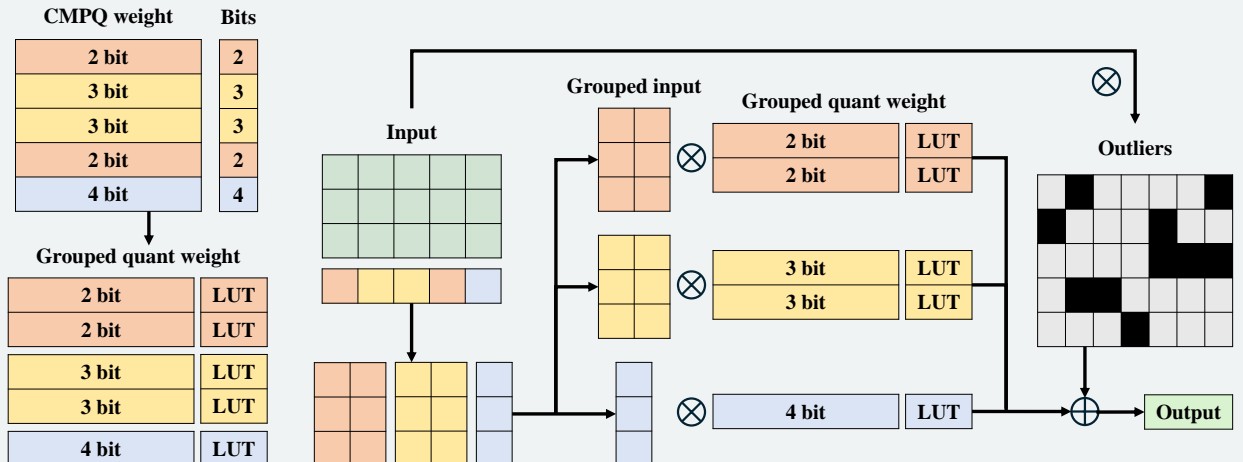

Figure 4: Framework of real quantization process of CMPQ. Given FP16 inputs and quantized weights, the features and weights are decomposed into sub-matrices based on the channel-wise precision allocation returned by Algorithm 1. Then each pair of feature and weight sub-matrices are multiplied together. The outlier matrices are multiplied in FP16 with the original input. Finally, both outlier and regular outputs are accumulated in 16-bit floating point outputs.

## 5.2 Time and Memory Usage for Quantization Process

In Table 8, we report the time and memory usage for two key procedures in CMPQ: (i) collecting layer activations and (ii) performing K-means clustering. The time required for collecting layer activations is less than 1 minute, and the peak memory usage of CMPQ is comparable to standard LLM inference. This is because CMPQ only requires forward propagation and does not need to compute the Fisher information matrix, as in SqueezeLLM. K-means clustering for the 7B model takes approximately 11 minutes, making CMPQ's computational time comparable to that of SqueezeLLM and GPTQ. These results indicate that CMPQ can be efficiently run on GPUs, with its activation collection process incurring minimal time and memory overhead.

Table 8: Peak memory requirement and time cost when quantizing different LLaMA2 models.

| Model | Running Time (min) | | | | Peak Memory (GB) | | |
|---|---|---|---|---|---|---|---|
| | Activation Collection | K-means | SqueezeLLM | GPTQ | CMPQ | SqueezeLLM | LLM (FP16) |
| LLaMA2-7B | 0.6 | 11 | 11.3 | 10 | 24.2 | 33 | 24.6 |
| LLaMA2-13B | 0.8 | 17 | 17.6 | 21 | 42.6 | 61 | 47.9 |

Table 9: Hardware profiling of latency and memory usage for 3-bit quantized LLaMA2-7B and LLaMA2-13B models, during the generation of 128 tokens on an A6000 GPU. Results are compared with SqueezeLLM using the standard transformer kernel implementation.

| Method | Latency (Seconds) | | Peak Memory (GB) | |
|---|---|---|---|---|
| | 7b | 13B | 7b | 13B |
| **SqueezeLLM** | 3.9 | 6.2 | 3.2 | 5.8 |
| **CMPQ w/o Outlier** | 3.84 | 6.84 | 3.58 | 6.44 |
| **CMPQ** | 3.96 | 6.91 | 3.59 | 6.44 |

## 6 Latency and Peak Memory Usage of CMPQ

We measure the latency and peak memory usage for generating 128 tokens on an A6000 GPU using the Torch CUDA profiler. For single-batch inference, we adopt the standard kernel implementation from SqueezeLLM (Kim et al., 2024), based on the widely used open-source codebase GPTQ-For-LLaMA. The quantization process is illustrated in Figure 4. Inspired by LLM.int8() (Dettmers et al., 2022), we handle the multiplication of matrices with different bit-widths separately. Outliers in the weight matrix are stored as CSR sparse matrices, with computations performed using a standard CSR-based kernel. Latency comparisons with SqueezeLLM were conducted on LLaMA2 models using 3-bit quantization, with the results presented in Table 9.

The results show that CMPQ preserves a similar deployment profile to integer-only quantization for the 7B model, while introducing a modest overhead for the 13B model. In particular, for 13B we observe an increase in both latency and peak memory (around 11–12%). This scaling trend is expected because CMPQ combines the dominant low-bit matrix multiplication with an auxiliary higher-precision path for protected weights and mixed-precision metadata. Although this auxiliary path is sparse, it does not benefit from low-bit acceleration to the same extent as the bulk quantized computation. As model size increases and inference becomes more memory-sensitive, the relative impact of this mixed-precision handling becomes more visible. Nevertheless, the dominant cost still comes from the low-bit computation, so CMPQ does not change the overall inference regime. Overall, CMPQ is designed to trade a small runtime/memory overhead for improved quantization quality and flexible adaptation to deployment-specific bit budgets.

## 7 Conclusion

In this work, we focused on mixed-precision quantization and aimed to design an algorithm capable of adapting to any bit-width constraint. We observed that different weight channels had varying impacts on model performance, and that activation distributions helped identify salient channels. Building on these insights, we proposed CMPQ, which integrated a channel-wise non-uniform quantization strategy. To further enhance performance, CMPQ introduced two types of outliers that collaboratively preserved critical information. Experimental results showed that CMPQ harnessed the potential of mixed-precision quantization in two key ways: (1) it achieved superior performance in integer-bit quantization tasks, and (2) it delivered significant performance improvements with only a modest increase in memory requirements. Besides, our latency analysis shows that CMPQ achieves comparable latency and memory usage to the integer-based quantization method.

**Acknowledgments**

This work is supported in part by the National Science Foundation (NSF) under grants CPS-2313110, ECCS-2143559, IIS-2006844, IIS-2144209, IIS-2223769, CNS-2154962, BCS-2228534, and CMMI-2411248; the Office of Naval Research (ONR) under grant N000142412636; the Commonwealth Cyber Initiative (CCI) under grant VVIQ24-011; and a research grant from Kneron, Inc.

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

# A Ablation Studies

## A.1 Impact of Sparsity Levels of CMPQ

In this section, we evaluate the trade-off between performance and outlier protection ratio, we adjusted the ratio of activation-based outliers from 0.05% to 0.45% and presented the perplexity results of the 3-bit quantized OPT-6.7B model on the C4 benchmarks, with varying outlier extraction percentages ranging from 0% to 0.5%, as shown in Figure 5. Notably, while we maintain a fixed protection ratio of 0.5% for quantization-based outliers across all experiments to ensure fair comparisons, the plot reveals that the perplexity gains diminish when the protection ratio exceeds 0.2%. This finding highlights CMPQ's potential to achieve superior performance with reduced storage requirements

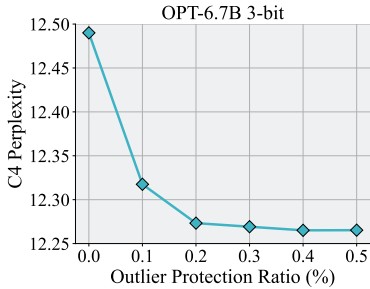

Figure 5: Outlier protection ratio and perplexity trade-off of 3-bit quantized OPT-6.7B model.

## A.2 Impact of Non-uniform Quantization

In Table 10, we provide a detailed analysis to further clarify the impact of non-uniform quantization. For uniform quantization, we apply the widely used round-to-nearest method with a group size of 128 for channel-wise weight quantization, while preserving 0.5% of activation-based outliers to ensure a fair comparison. Additionally, we report the best perplexity achieved by the baseline methods. As shown in Table 10, across various bit-widths and model sizes, non-uniform quantization consistently outperforms uniform quantization, particularly in extremely low-bit (2-bit) settings. This is because the non-uniform distribution of weights leads to inefficient utilization of the quantization bins in uniform quantization, where some bins may remain underutilized or unused. Interestingly, we also observe that for certain tasks, uniform quantization can improve perplexity (e.g., OPT-13B at 3-bit on WikiText2). In such cases, equipping CMPQ with uniform quantization yields the best performance.

Table 10: Perplexity ($\downarrow$) comparison on the C4 and WikiText-2 datasets. LLMs are quantized by CMPQ using non-uniform and uniform approaches.

| Method | Avg. Bit | OPT-2.7B | | OPT-6.7B | | OPT-13B | |
| --- | --- | --- | --- | --- | --- | --- | --- |
| | | Wiki | C4 | Wiki | C4 | Wiki | C4 |
| Baseline | 2 | >100 | 38.07 | 22.33 | 21.62 | **16.02** | 16.60 |
| Uniform | 2 | >100 | 80.52 | >100 | >100 | >100 | >100 |
| Non-Uniform | 2 | **32.46** | **28.32** | **18.63** | **18.31** | 16.48 | **16.30** |
| Baseline | 3 | 16.32 | 15.63 | 11.41 | 12.30 | 10.50 | 12.39 |
| Uniform | 3 | 13.49 | 14.06 | **11.41** | 12.28 | **10.42** | 11.65 |
| Non-Uniform | 3 | **13.38** | **14.05** | 11.45 | **12.26** | 10.67 | **11.64** |
| Baseline | 4 | 12.69 | 13.48 | 10.93 | 11.86 | 10.21 | 11.28 |
| Uniform | 4 | 12.63 | 13.34 | **10.91** | 11.84 | 10.19 | 11.27 |
| Non-Uniform | 4 | **12.63** | **13.33** | 10.95 | **11.83** | **10.17** | **11.27** |

## A.3 Data Efficiency for the Calibration Set

In Table 11, we present a data efficiency analysis based on the number of data samples in the calibration datasets and compare the perplexity of the LLaMA2-7B model under 3-bit quantization. Although a calibration set of 128 data samples is used consistently throughout the paper, our method typically achieves the desired quantization performance with as few as single-digit sample sizes. This efficiency stems from the fact that we do not rely on regression or backpropagation; instead, we only measure the activation norm from the calibration set, making the process highly data-efficient. In contrast, both GPTQ and AWQ require more than 50 data points for calibration, as reported in Kim et al. (2024).

Table 11: Data efficiency analysis of calibration datasets: Comparing CMPQ and SqueezeLLM for perplexity on C4 and WikiText-2 with 3-bit quantization of the LLaMA2-7B across varying calibration set sizes.

| Methods | Tasks | Nmuber of calibration samples | | | | | |
|---------|-------|------|------|------|------|------|------|
| | | 1 | 2 | 5 | 10 | 20 | 100 |
| **SqueezeLLM** | Wiki | 6.41 | 6.22 | 6.20 | 6.16 | 6.16 | 6.18 |
| | C4 | 7.89 | 7.81 | 7.73 | 7.72 | 7.72 | 7.72 |
| **CMPQ** | Wiki | 6.18 | 6.14 | 6.16 | 6.14 | 6.14 | 6.14 |
| | C4 | 7.71 | 7.65 | 7.66 | 7.65 | 7.65 | 7.66 |

# B  Additional Experimental Results

## B.1  Additional Experimental Results on More LLMs

In Table 12, we present a comparison of quantization results across additional LLMs, including models from the OPT family ranging from 1.3B to 30B parameters, LLaMA2-70B, as well as the more recent LLaMA3-7B. Our findings are consistent with the main results, indicating that CMPQ generally outperforms all baselines. We also note that given the block-wise quantization nature (i.e., 128 weights share one scale value), additional bits allocated by such methods (AWQ, GPTQ, and QuIP) could exceed those introduced by CMPQ as the model size grows, which further strength the performance of our proposed method.

Table 12: Additional perplexity (↓) comparison of {2, 3, 4}-bit quantized LLMs on the C4 and WikiText-2 datasets. Bold font indicates the best performance across all methods, while underlined results denote the second-best.

| Method | Avg. Bit | OPT-1.3B | | OPT-13B | | OPT-30B | | LLaMA2-70B | | LLaMA3-8B | |
|--------|----------|-------|-------|-------|-------|-------|-------|-------|-------|-------|-------|
| | | Wiki | C4 | Wiki | C4 | Wiki | C4 | Wiki | C4 | Wiki | C4 |
| **FP16** | 16 | 14.62 | 14.72 | 10.13 | 11.2 | 9.56 | 10.69 | 3.32 | 5.52 | 6.1 | 9.2 |
| **RTN** | 2 | >1000 | >1000 | >1000 | >1000 | >1000 | >1000 | – | – | >1000 | >1000 |
| **GPTQ** | 2 | >1000 | >1000 | 372.68 | 135.48 | 71.7 | 29.59 | – | – | 210 | >1000 |
| **AWQ** | 2 | – | – | – | – | – | – | – | – | >1000 | >1000 |
| **QuIP** | 2 | 41.64 | **29.78** | **16.02** | 16.6 | **11.48** | 13.55 | – | – | **85.1** | 130 |
| **CMPQ** | 2 | **41.58** | 36.67 | 16.48 | **16.3** | 11.53 | **12.89** | – | – | 120 | **110** |
| **RTN** | 3 | >1000 | >1000 | >1000 | >1000 | >1000 | >1000 | 7.52 | 10.02 | 27.9 | 110 |
| **GPTQ** | 3 | 21.35 | 21.59 | 11.6 | 13.34 | 10.32 | 12.23 | 4.86 | 6.69 | 8.2 | 13.7 |
| **AWQ** | 3 | 16.32 | 16.28 | 10.67 | 12.61 | 9.85 | **10.96** | 3.74 | 5.81 | 8.2 | 11.6 |
| **QuIP** | 3 | 16.21 | 17.12 | **10.5** | 12.39 | **9.79** | 11.66 | 3.85 | 6.14 | **7.5** | 11.3 |
| **CMPQ** | 3 | **16.07** | **16.08** | 10.67 | **11.64** | 9.83 | 10.98 | **3.70** | **5.78** | 7.89 | **11.3** |
| **RTN** | 4 | 47.62 | 27.2 | 11.32 | 12.35 | 10.77 | 13.52 | 3.67 | 5.80 | 8.5 | 13.4 |
| **GPTQ** | 4 | 15.59 | 16.96 | 10.31 | 12.26 | 9.63 | 11.8 | 3.59 | 5.70 | 7.0 | 10.4 |
| **AWQ** | 4 | 14.94 | 15.04 | 10.22 | 11.28 | **9.59** | 10.75 | 3.41 | 5.58 | 6.6 | 9.4 |
| **QuIP** | 4 | 14.88 | 16.38 | 10.21 | 12.16 | 9.61 | 11.5 | 3.53 | 5.86 | 6.6 | 11.3 |
| **CMPQ** | 4 | **14.84** | **14.99** | **10.17** | **11.27** | 9.61 | **10.74** | **3.39** | **5.57** | **6.5** | **9.39** |

## B.2  Visualization of channel-wise bit allocation

To better understand the behavior of CMPQ, we visualize the bit allocation for the `out proj` weight matrices across all 32 layers of LLaMA2-7B under an average bit budget of 3.4. Since each matrix contains 4096 channels, we partition channels into 64 contiguous bins and report the average assigned bit-width within each bin, while preserving the original channel ordering. The resulting heatmap is shown in Figure 6.

Several observations can be made. First, the learned allocation is clearly non-uniform within each layer, indicating that CMPQ does not assign precision uniformly even for channels belonging to the same weight matrix. This supports our motivation that channel sensitivities are heterogeneous and that layer-wise uniform assignment may be too coarse. Second, even under an average budget of 3.4 bits, the higher-precision assignments remain concentrated on only a subset of channel groups, while the majority of channels stay close to

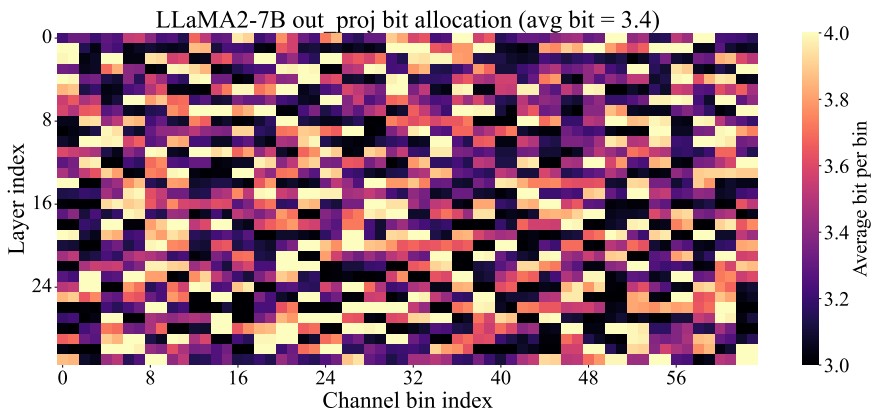

Figure 6: Channel-wise bit allocation for the out projection layer weight matrices across all 32 layers of LLaMA2-7B under an average bit budget of 3.4. CMPQ produces non-uniform allocation both within and across layers, indicating heterogeneous channel sensitivity under a fixed memory budget.

the base precision. This indicates that CMPQ uses the additional budget selectively rather than uniformly increasing precision everywhere.

Overall, this visualization provides qualitative evidence that CMPQ assigns a structured mixed-precision pattern across the network. Rather than treating each layer as a single unit, CMPQ distributes precision at the channel level, enabling finer control of quantization error under a fixed memory budget.

### B.3 Justification for Activation-Based Precision Allocation

Our mixed-precision allocation strategy assigns higher precision to channels with larger activation magnitudes. Although this design is heuristic, it is motivated by the role of activations in weight-only post-training quantization. For a linear projection $Y = XW$, the output perturbation caused by weight quantization can be written as

$$\Delta Y = X(W - \widehat{W}), \tag{1}$$

where $\widehat{W}$ denotes the quantized weight matrix. Thus, the impact of a weight quantization error is modulated by the corresponding input activations. Channels that are more strongly activated tend to have a larger influence on the output, and assigning them overly low precision may introduce disproportionately large output distortion. This motivates using the channel-wise activation norm as a practical proxy for quantization sensitivity.

We use the channel-wise activation $\ell_2$ norm as the default allocation criterion because it provides a simple and lightweight estimate of channel importance. Unlike gradient-based or reconstruction-based criteria, it can be obtained from a single calibration pass without retraining, gradient computation, or repeated provisional quantization. This makes it particularly suitable for post-training quantization, where efficiency and ease of deployment are important. The use of activation statistics is also consistent with prior observations in activation-aware quantization methods such as AWQ (Lin et al., 2024), which shows that activation-aware saliency is more informative than weight magnitude alone for identifying important weights. In contrast, Fisher-information-based sensitivity, as used in methods such as SqueezeLLM (Kim et al., 2024), can provide a meaningful importance signal but requires additional gradient-based second-order approximation, making it more costly in our setting.

To further validate this design choice, we compare the activation $\ell_2$ norm with closely related alternatives, including the activation $\ell_1$ norm and $\ell_\infty$ norm. Table 13 reports the perplexity of OPT models under 3-bit quantization. Across both OPT-2.7B and OPT-6.7B, the $\ell_2$ norm consistently achieves the best performance, suggesting that it is a more effective criterion for identifying salient channels and guiding precision allocation.

Table 13: Comparison of different activation-norm criteria for precision allocation under 3-bit quantization. Lower perplexity is better.

| Model | $\ell_\infty$ | $\ell_1$ | $\ell_2$ |
|-------|------|------|------|
| OPT-2.7B | 15.48 | 14.67 | **14.05** |
| OPT-6.7B | 14.76 | 13.23 | **12.26** |

Table 14: Quantization results of SliM-LLM and CMPQ on WikiText-2 dataset.

| Method | Avg.Bit | LLaMA2-7B | LLaMA2-13B |
|--------|---------|-----------|------------|
| SliM-LLM | 2 | 20.00 | 12.14 |
| CMPQ | 2 | **14.37** | **9.14** |
| SliM-LLM | 3 | 7.74 | 6.26 |
| CMPQ | 3 | **6.14** | **5.34** |

We also consider other possible allocation strategies. Weight-magnitude-based allocation does not account for how frequently or strongly a channel is activated, and therefore may fail to identify channels that are important for the model's actual computation. Fisher-information-based allocation captures sensitivity more directly but introduces gradient estimation and additional preprocessing cost. Reconstruction-error-based allocation is also more expensive because it requires evaluating provisional quantization results under different allocation choices. Therefore, activation $\ell_2$ norm offers a favorable balance between effectiveness, simplicity, and efficiency. Together with the empirical results above, this supports our choice of activation $\ell_2$ norm as a robust and practical criterion for mixed-precision allocation.

### B.4 Fractional Bit Quantization on WikiText-2

To complement the C4 results in the main text, we report here the corresponding comparison between CMPQ and LLM-MQ under fractional bit-width constraints on WikiText-2. The results exhibit the same overall trend as in Figure 3: CMPQ consistently outperforms LLM-MQ across different average bit-widths and model sizes, with the largest advantage appearing in the low-bit regime. In particular, the results on WikiText-2 in Figure 7 further support our main observation that channel-wise mixed-precision allocation makes more effective use of fractional-bit budgets than layer-wise allocation, especially when the quantization budget is highly constrained.

## C Comparison with Other Weight-only Quantization Methods

To broaden the experimental coverage, we further include comparisons with one recent mixed-precision method, SliM-LLM (Huang et al., 2024b), as well as three state-of-the-art PTQ approaches: SqueezeLLM (Kim et al., 2024), GPTVQ (Van Baalen et al., 2024), and VPTQ (Liu et al., 2024a). Note that SqueezeLLM (Kim et al., 2024) also quantizes LLMs in a channel-wise way, but it depends on back-propagation.

### C.1 Comparison with SliM-LLM

We present a quantitative comparison between CMPQ and SliM-LLM (Huang et al., 2024b) on the LLaMA2 model family. For a fair evaluation, both methods are calibrated using 128 randomly selected 2048-token segments from the C4 dataset. The quantized models are then evaluated based on perplexity scores on the WikiText2 dataset. As shown in Table 14, CMPQ consistently outperforms SliM-LLM on both LLaMA2-7B and 13B models at 3-bit and 2-bit precision. The performance gain stems from CMPQ's more sophisticated bit allocation strategy, using channel-wise precision assignment rather than SliM-LLM's group-wise approach, as well as its two dedicated outlier protection mechanisms. Furthermore, while SliM-LLM is primarily designed for integer-bit quantization, CMPQ tackles a more challenging and practical objective: enabling

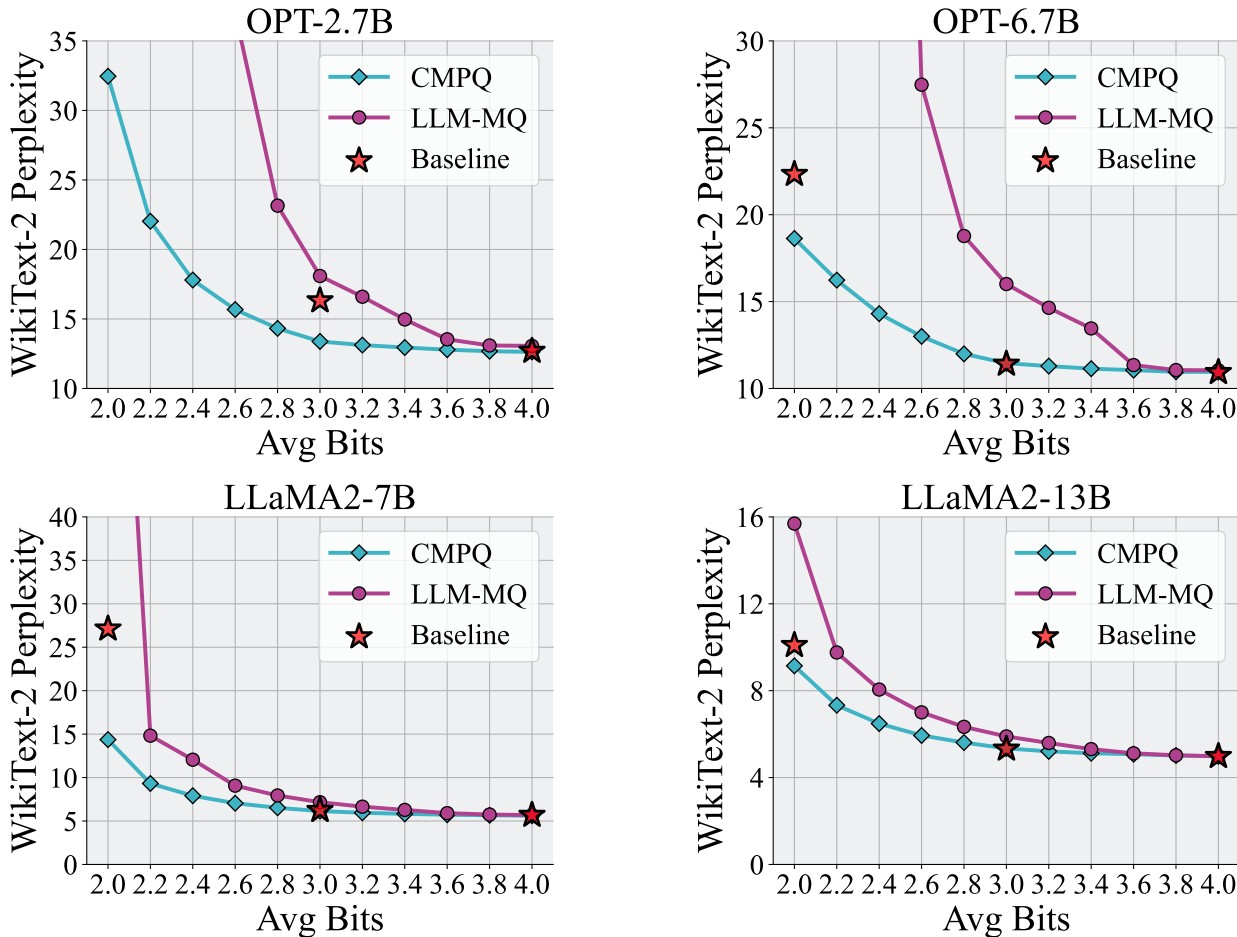

Figure 7: Comparison of WikiText-2 perplexity between CMPQ and LLM-MQ for fractional bit-width quantization. The red star indicates the best performance achieved by integer-based baselines at {2, 3, and 4} bits.

Table 15: Quantization results of SqueezeLLM (SqzLLM) and CMPQ on the WikiText-2 dataset. The Memory column includes the memory requirements for inference (needed by CMPQ) and memory requirements for backpropagation with a batch size of 1 (needed by SqueezeLLM). For CMPQ, we also report performance at 2.2-bit and 3.2-bit to facilitate trade-off discussions.

| Model | Memory (GB) | SqzLLM 2 | CMPQ 2/2.2 | SqzLLM 3 | CMPQ 3/3.2 |
|---|---|---|---|---|---|
| **OPT-2.7B** | 9.88/19.76 | – | 32.46/**22.03** | 13.43 | 13.38/**13.12** |
| **OPT-6.7B** | 24.8/49.61 | – | 18.63/**16.23** | 11.31 | 11.45/**11.29** |
| **LLaMA2-7B** | 24.61/49.23 | 10.79 | 14.37/**9.32** | 5.96 | 6.14/**5.95** |
| **LLaMA2-13B** | 47.88/95.76 | 7.91 | 9.14/**7.33** | 5.23 | 5.34/**5.21** |

highly adaptable, fine-grained post-training quantization with continuous bit allocation across channels, making it more suitable for real-world deployment.

## C.2 Comparison with SqueezeLLM

While we primarily compare with baselines that do not rely on backpropagation, we acknowledge that gradient information, at the cost of extra resources, can indeed enhance the performance of quantized LLMs. In Table 15, we compare our method with SqueezeLLM (Kim et al., 2024), which also employs non-uniform quantization but uses gradient information to weight the clustering process, safeguarding more sensitive weights. Additionally, we report the memory requirements for loading LLMs and performing backpropagation in FP16 precision[1].

As shown in Table 15, though SqueezeLLM slightly outperforms CMPQ across various models at different quantization levels, this *mild* improvement comes at a significant cost: SqueezeLLM requires **two times** the memory for the quantization process, making it impractical for larger models, especially under resource constraints. In contrast, CMPQ offers a more efficient solution when memory is limited, requiring only $1/2$ of the computational resources. Moreover, if an additional 10% of storage space is available, CMPQ can achieve better performance than SqueezeLLM, particularly in low-memory environments. At higher precision, such as 4-bit quantization, the tradeoff between computational resource requirements and model storage is less pronounced. The maximum performance gain of SqueezeLLM over CMPQ is marginal, only $(5.61 - 5.57)/5.61 = 0.71\%$. This minimal improvement renders the substantial additional resource demands of SqueezeLLM unnecessary.

Overall, CMPQ not only provides comparable performance with SqueezeLLM in the integer-only quantization tasks, but the method also offers significant advantages by eliminating the 100% increase in computational overhead associated with SqueezeLLM's gradient-based approach, with only a 10% increase in storage. Furthermore, CMPQ is more versatile, as it applies to non-integer bit quantization tasks, making it a more practical option for a wider range of scenarios.

## D  Limitations

- **Weight-only quantization.** The present work focuses on weight-only post-training quantization and does not consider activation quantization. While this design choice simplifies deployment and isolates the effect of channel-wise mixed precision on weight compression, it does not capture the additional benefits or challenges of joint weight–activation quantization.

- **Discrete precision set.** The proposed method supports arbitrary average bit-widths by mixing a limited set of discrete precision levels. Accordingly, its flexibility arises from adaptive allocation over this discrete set {2,3,4}, rather than from optimization over a fully continuous or unrestricted precision space.

- **Preprocessing scalability.** CMPQ relies on channel-wise K-means quantization, which introduces offline preprocessing overhead. Although this overhead does not affect inference-time efficiency, it may scale less favorably for extremely large models or repeated retargeting to many deployment budgets.

- **Calibration-set representativeness.** The channel allocation strategy depends on activation L2-norms estimated from a calibration set. Therefore, the method implicitly assumes that the calibration data provides a representative estimate of channel importance under the target usage distribution. Performance may degrade when this assumption is violated.

- **Evaluation regime.** Our study is centered on the practically important low-bit setting for LLM deployment, especially the 2–4 bit range. As a consequence, the current paper does not provide an extensive investigation of behavior outside this regime.

---

[1]The memory requirements were calculated using the Hugging Face platform.

