# OpenReview forum: "Adapting to Any Bit-Width: Channel-Wise Mixed-Precision Quantization for LLMs"
_TMLR — Accepted by TMLR_

### Review · Reviewer_mqUX · 2026-02-23

**Summary Of Contributions:**

This paper propose a method that allocate non-uniform quantization level to different channels. Some channels with smaller activation norm are assigned lower quantization precision.

**Additional Comments:**

See requested change. For TMLR, novelty is not important in determining the acceptance of  a paper. However, this paper looks very incremental in terms of scientific contribution and therefore is less interesting to read.

**Audience:**

Yes

**Audience Explanation:**

Some may be interested.

**Claims And Evidence:**

Yes

**Claims Explanation:**

Yes, comprehensive experiments are done to justify that extending channel wise quantization level is benficial and outperform layer-wise. But the claim "non-uniform channel-wise quantization level is more beneficial than non-uniform layer-wise quantization" may not be universally true.

**Requested Changes:**

1. The novelty of the proposed method are limited. Existing work already use activation norm as indicator to decide the quantization level  for each layer. This work simply extend the same indicator to channel-wise quantization level. There are no particular new findings that are of interest. This is not a requested change but I just want to express this as the main weakness of the paper for the AE to consider as well.

2. An important claim of this paper is that channel-wise quantization level outperform layer-wise quantization level. However, except benchmark result (i.e., accuracy), there is no sufficient claim (e.g., other statistical result) that explain why channel-wise quantization level is more beneficial. Without a theory (or reasonable justification), this conclusion may not be rigorous. For example, training hyper-parameters might have very big impact of whether this claim is universially true.   I suggest the authors to do more study on this aspect.

I will take how the authors address the second weakness into account when I do the final evaluation.

---

> ### Author Response · Authors · 2026-03-10
> **Response from the Authors**
>
> >**1.** The novelty of the proposed method are limited. Existing work already use activation norm as indicator to decide the quantization level for each layer. This work simply extend the same indicator to channel-wise quantization level. There are no particular new findings that are of interest. This is not a requested change but I just want to express this as the main weakness of the paper for the AE to consider as well.
> >
> **Response**:
> We thank the reviewer for raising this concern. While CMPQ builds on ideas used in prior work (e.g., activation-based importance estimation), the novelty of our approach primarily lies in its **problem formulation and design objective.**
>
> Most existing post-training quantization methods are formulated around fixed integer-bit settings (e.g., 2-bit, 3-bit, or 4-bit) and focus on optimizing accuracy under these discrete configurations. In contrast, CMPQ formulates quantization as a flexible mixed-precision allocation problem under an arbitrary average bit budget. This formulation allows the model to operate at fractional bit-widths (e.g., 2.5 or 3.3 bits) by distributing precision across channels.
>
> This shift in formulation naturally leads to a channel-wise precision allocation strategy, which enables finer control over quantization error compared to layer-wise schemes. It also allows CMPQ to adapt smoothly to deployment constraints where the available memory budget does not correspond to a standard integer bit-width.
>
> In addition, prior approaches typically utilize additional storage budget by protecting more weights in FP16, resulting in a stepwise and often diminishing improvement as the number of protected weights increases (as discussed in AWQ). In contrast, CMPQ actively redistributes precision across low-bit channels while keeping the FP16 footprint fixed, enabling more effective use of the available memory budget without increasing inference overhead.
>
> Overall, CMPQ introduces a different formulation of the quantization problem: optimizing channel-wise precision allocation under arbitrary bit budgets, which leads to a practical and flexible quantization framework for real-world deployment.
>
>
> >**2.** An important claim of this paper is that channel-wise quantization level outperform layer-wise quantization level. However, except benchmark result (i.e., accuracy), there is no sufficient claim (e.g., other statistical result) that explain why channel-wise quantization level is more beneficial. Without a theory (or reasonable justification), this conclusion may not be rigorous. For example, training hyper-parameters might have very big impact of whether this claim is universially true. I suggest the authors to do more study on this aspect.
> >
> **Response**:
> We thank the reviewer for this valuable suggestion. We agree that providing additional intuition for why channel-wise precision allocation is beneficial would strengthen the paper.
>
> Our motivation is based on the empirical observation that **quantization sensitivity varies significantly across channels within the same layer**. In large language models, different channels often capture distinct features and exhibit very different activation magnitudes. As a result, their tolerance to quantization noise can vary considerably. When a single bit-width is assigned to an entire layer, all channels must share the same precision. This can lead to **inefficient use** of the bit budget: insensitive channels may receive more precision than necessary, while highly sensitive channels may suffer from excessive quantization error. In contrast, channel-wise allocation allows precision to be distributed according to channel importance, enabling more effective control of quantization error within each layer.
>
> Importantly, this behavior is **largely independent** of training hyperparameters, since our method operates in a **post-training quantization setting** and relies *only* on activation statistics collected from the pretrained model. These statistics reflect the intrinsic behavior of the model rather than training dynamics.
>
> We agree that deeper theoretical analysis would be valuable. Similar to many prior mixed-precision quantization works, our approach is primarily empirically motivated, and our experiments consistently demonstrate that channel-wise allocation improves performance under the same bit budget across multiple models and settings.

---

### Review · Reviewer_J3sN · 2026-02-25

**Summary Of Contributions:**

This paper proposes **CMPQ**, a post‑training **channel‑wise mixed‑precision quantization** framework for LLMs that supports **arbitrary (including fractional) average bit‑widths**. The method combines:

- **Channel‑wise precision allocation** based on activation L2 norms

- **Non‑uniform quantization** via K‑means clustering

- **Dual strategy** **for** **outlier identification** (activation‑based + quantization‑aware)


Experiments across two LLM families (OPT and LLaMA2) show that CMPQ:

- Outperforms integer‑bit PTQ baselines (GPTQ, AWQ, QuIP, SqueezeLLM, etc.)

- Outperforms the mixed‑precision baseline LLM‑MQ

- Achieves strong results even at **f**ractional bit‑widths, offering smooth memory–accuracy trade‑offs

- Maintains competitive latency and memory usage


**Strengths**

- Clear motivation: integer‑bit quantization wastes storage flexibility; fractional bits are underexplored.

- Methodologically simple yet effective; avoids heavy optimization or backprop.

- Fractional‑bit evaluation is novel and practically relevant.

- Good robustness to calibration data choice.


**Weaknesses**

- Some design choices (e.g., 0.45% + 0.05% outlier ratios, 1% 4‑bit protection at 3‑bit) appear heuristic.

- Limited ablations on the interaction between non‑uniform quantization and outlier extraction.

- Efficiency claims could be strengthened with more detailed latency breakdowns.

**Audience:**

Yes

**Audience Explanation:**

LLM quantization is a highly active research area, and fractional‑bit quantization is underexplored but practically important. The paper fits well within TMLR’s scope.

**Broader Impact Concerns:**

The paper focuses on model compression and does not introduce new ethical risks beyond those inherent to LLM deployment.

**Claims And Evidence:**

Yes

**Claims Explanation:**

The empirical evidence is extensive and convincing:

- Evaluations span multiple model families and sizes (2.7B → 70B).

- Comparisons include all major PTQ baselines.

- Fractional‑bit results clearly demonstrate CMPQ’s advantage over LLM‑MQ.

- Ablations (e.g., preliminary study, outlier types) support the design choices.

- Latency and memory profiling show CMPQ is competitive with existing low‑bit methods.


However, some methodological choices rely on empirical heuristics without deeper analysis. While not fatal, more justification would strengthen the paper.

**Requested Changes:**

### Critical (needed for acceptance)

1. **Clarify the rationale behind the specific outlier ratios (0.45% + 0.05%).** Provide sensitivity analysis or justification for why these values generalize across models.

2. **Provide more ablation on the interaction between non‑uniform quantization and outlier extraction.** For example: performance with only activation‑based outliers, only quantization‑aware outliers, or neither.

3. **Improve clarity on the computational cost of CMPQ.**  A more detailed breakdown (e.g., K‑means cost per layer, calibration pass time) would help readers assess deployability.


### Non‑critical (would strengthen the paper)

4. **Add more downstream task results in the main paper.**  Perplexity is useful but not always predictive of real‑world performance. Reasoning and agentic tasks could be considered.

5. **Provide qualitative examples or visualizations** of channel‑wise bit allocation patterns across layers.

---

> ### Author Response · Authors · 2026-03-10
> **Response from the Authors (1)**
>
> ## Critical Changes
>
> >**1.** Some design choices appear heuristic.
> >
> **Response**: We thank the reviewer for raising this question. Our design choices are based on two considerations: **fair comparison** and **the channel-wise behavior observed in our method**.
>
> First, we keep the **total FP16 protection budget at 0.5%** to remain directly comparable with prior outlier-aware PTQ methods, like LLM-MQ and SqueezeLLM. Using the same overall budget therefore allows us to isolate the effect of our main contribution, **channel-wise mixed-precision allocation**, rather than gaining accuracy by increasing FP16 storage.
>
> Second, the choice of 0.45% activation-based outliers + 0.05% quantization-aware outliers is **not arbitrary**. In particular, SqueezeLLM reports that preserving 0.45% outliers together with 0.05% highly sensitive weights substantially improves quantization performance. At the same time, CMPQ **does not** use this budget in the same way. As shown in Fig. 2 and discussed in Sec. 3.2.2, we observe that outliers in LLMs exhibit a clear channel-wise pattern: if a channel is salient, it tends to remain salient across tokens. This motivates allocating the larger portion (0.45%) to activation-based outliers, identified from the top values of the channel-wise activation L2 norm. The remaining 0.05% is then used for quantization-aware outliers, i.e., weights with the largest post-quantization errors, which specifically compensates for weights that distort the K-means centroids and are not fully captured by activation statistics alone. Thus, **the budget is grounded in prior empirical evidence, but the allocation criterion and the role it plays in CMPQ are specific to our method.**
>
> Regarding the reviewer’s question on the **1% 4-bit protection in the 3-bit setting**, this is also supported by our preliminary study in Table 1. There, we show that naive mixed-precision schemes that compensate higher-precision channels with equally many 2-bit channels are not effective: extending mixed precision to too many channels degrades performance, and the paper attributes this to the fact that the information loss from 2-bit quantization can outweigh the gains from 4-bit quantization. In fact, 1% 2b + 1% 4b is **better** than 10% 2b + 10% 4b, and is roughly on par with or slightly better than uniform 3-bit, which led us to protect only a **small fraction of salient channels** at 4-bit rather than aggressively introducing more 2-bit channels. This design is therefore tied to our empirical finding that **very small, targeted high-precision protection is beneficial, while broader low-bit compensation can be harmful**.
>
> More broadly, these design choices reflect the empirical observation that only a very small fraction of weights dominate quantization error in large language models. Allocating slightly higher precision to this small subset is therefore more effective than aggressively reducing the precision of other weights. Importantly, CMPQ keeps this protection budget fixed across all evaluated models and datasets and still achieves consistent improvements, suggesting that these ratios generalize well in practice. We will revise the paper to make this rationale clearer: the **overall budget** follows prior work for comparability, but the **allocation strategy and criteria** are specific to CMPQ and are motivated by our channel-wise observations and preliminary analysis.
>
> >**2.** Limited ablations on the interaction between non‑uniform quantization and outlier extraction.
> >
> **Response**: We thank the reviewer for this helpful suggestion. We would like to clarify that this ablation is **already included in Table 5** of the paper. Specifically, we report results for (i) no outlier protection, (ii) activation-based outliers only, (iii) quantization-aware outliers only, and (iv) both together, which directly addresses the interaction between the two components. For a consistent comparison, when one type of outlier is removed, **we allocate the full 0.5% FP16 budget to the remaining type**, so that all settings are compared under the same storage constraint.
>
> The results in Table 5 show that both mechanisms are beneficial and that they play complementary roles. Removing all outlier protection leads to the largest degradation. Using only activation-based outliers or only quantization-aware outliers both improve performance, while using both generally gives the best overall results. This is consistent with the intended design: activation-based outliers protect salient channels identified from activation statistics, whereas quantization-aware outliers protect weights that would otherwise induce large quantization error and distort the non-uniform clustering process.

---

> ### Author Response · Authors · 2026-03-10
> **Response from the Authors (2)**
>
> >**3.** Efficiency claims could be strengthened with more detailed latency breakdowns. Improve clarity on the computational cost of CMPQ.  A more detailed breakdown (e.g., K‑means cost per layer, calibration pass time) would help readers assess deployability.
> >
> **Response**: We thank the reviewer for this helpful suggestion. We agree that clarifying the computational cost of CMPQ would make the deployment setting easier to assess. We would like to note that the paper already reports the two main preprocessing costs of CMPQ **in Table 8**: activation collection and K-means clustering. **The activation collection corresponds to the calibration forward pass**, which takes 0.6 min for LLaMA2-7B and 0.8 min for LLaMA2-13B. The dominant offline cost is K-means clustering, which takes 11 min and 17 min, respectively. Thus, the calibration pass is lightweight, while the main one-time preprocessing cost comes from quantization itself.
>
> To provide additional intuition, for LLaMA2-7B, the reported 11 min K-means time is the total cost for the full model, which corresponds to roughly **$11*60/32=20.6$ seconds per layer on average**. Since different matrices have different shapes (e.g., attention projections vs. MLP projections), this is only an approximate average rather than a true per-layer profile. We will clarify this point in the revision.
>
> Importantly, these costs are **offline model preparation costs**  and do not affect online inference latency. On the deployment side, Table 9 already shows that CMPQ achieves latency and peak memory comparable to SqueezeLLM, indicating that the sparse FP16 outliers introduce only marginal runtime overhead.
> We will revise the paper to more clearly separate quantization-time preprocessing cost from inference-time cost, and to explicitly state that K-means is the dominant one-time cost while the calibration pass itself is relatively small.
>
>
> ## Non-critical Changes
>
> >**4.** Add more downstream task results in the main paper.  Perplexity is useful but not always predictive of real‑world performance. Reasoning and agentic tasks could be considered.
> >
> **Response**: We thank the reviewer for this helpful suggestion. We agree that perplexity alone does not fully capture practical model quality. In fact, beyond perplexity, we already include additional downstream evaluations in **Section 4.7 and Table 6**, where we report results on PIQA, HellaSwag, and 5-shot MMLU. These benchmarks complement perplexity by covering commonsense reasoning and in-context learning ability, and CMPQ remains competitive on these tasks.
>
> In addition, in our response to `Reviewer v4PY Q8`, we further provide 4-bit results on GSM8K for Llama-2 7B/13B, which offer additional evidence on a math reasoning benchmark. Together, these results suggest that the benefits of CMPQ are not limited to language modeling metrics alone, but also transfer to downstream task performance.
>
> We agree that broader evaluations, including more agentic settings, would be valuable future work, but we believe the current results already provide meaningful evidence beyond perplexity alone.
>
> >**5.** Provide qualitative examples or visualizations of channel‑wise bit allocation patterns across layers.
> >
> **Response**: Thank you for this helpful suggestion. We agree that visualizing the channel-wise allocation can make CMPQ more intuitive. In the revised version, we add a visualization of the bit allocation pattern for the `out_proj` matrices across all 32 layers of LLaMA2-7B under an average budget of 3.4 bits (**Appendix B.2 and Figure 6**). Since each matrix contains thousands of channels, a raw channel-by-channel plot is difficult to read; therefore, we group neighboring channels into 64 bins and plot the average assigned bit-width in each bin while preserving the original channel ordering. This allows us to reveal the structure of the allocation without introducing artifacts from sorting. The resulting figure shows that CMPQ produces clearly non-uniform allocation within layers, with only a subset of channel groups receiving higher precision. We will include this visualization and discussion in the revised manuscript to better illustrate the behavior of the proposed channel-wise mixed-precision scheme.

---

### Review · Reviewer_v4PY · 2026-03-02

**Summary Of Contributions:**

**Summary**
This paper proposes CMPQ (Channel-Wise Mixed-Precision Quantization), a post-training quantization framework for LLMs that assigns different bit-widths (2, 3, or 4 bits) to individual weight channels based on the L2-norm of their corresponding activations. The method aims to support arbitrary average bit-widths, including fractional values (e.g., 2.4 or 3.8 bits), thereby allowing better utilization of available device storage compared to integer-only quantization schemes. CMPQ combines three main components: (1) channel-wise precision allocation guided by activation norms (Algorithm 1), (2) non-uniform quantization via K-means clustering per channel, and (3) a dual outlier protection strategy that separately handles activation-based outliers (0.45% of weights) and quantization-error-based outliers (0.05% of weights), both stored in FP16. Experiments are conducted on OPT and LLaMA2 families across WikiText-2 and C4 perplexity benchmarks, with additional results on commonsense QA and MMLU tasks. The authors report consistent improvements over baselines (RTN, GPTQ, AWQ, QuIP, LLM-MQ) particularly at 2-bit and 3-bit settings, and demonstrate that fractional-bit quantization can yield significant perplexity gains with modest storage increases.

**Key Strengths**: Practical problem formulation (fractional-bit adaptation); simple and efficient algorithm requiring only forward passes; thorough ablation studies and robustness analysis.

**Key Weaknesses**: Limited novelty in individual components; narrow experimental evaluation missing important recent baselines and tasks; questionable fairness of some comparisons; the precision allocation heuristic lacks theoretical grounding.

**Additional Comments:**

N/A

**Audience:**

Yes

**Audience Explanation:**

LLM quantization is an active and practically important research area with clear relevance to the TMLR audience. The specific problem of fractional-bit quantization for better device utilization is practically motivated and relatively underexplored. Practitioners deploying LLMs on heterogeneous hardware with varying memory budgets would find the concept of continuous bit-width adaptation useful. The paper's focus on post-training quantization without backpropagation also aligns with practical constraints.
That said, the audience interest is somewhat diminished by the limited scope of models tested (primarily OPT and LLaMA2, which are increasingly outdated), and the absence of comparisons with several important recent methods in the main text (VPTQ results are relegated to the appendix, and methods like QuIP# are referenced but not compared against).

**Broader Impact Concerns:**

This paper addresses model compression, which generally has positive societal implications by making LLMs more accessible on resource-constrained devices and reducing energy consumption during inference. There are no obvious direct ethical concerns specific to this work. The techniques are general-purpose and do not target any particular application domain that would raise red flags. A broader impact statement is not strictly required, though the authors could briefly note that enabling LLM deployment on edge devices carries both the benefits of democratized access and the risks of wider availability of potentially harmful model outputs.

**Claims And Evidence:**

No

**Claims Explanation:**

**Adapting to any bit-width** The method only mixes 2-bit, 3-bit, and 4-bit channels. This provides fractional averages only within the [2, 4] range and only at a coarse granularity. The title and framing overstate the flexibility — the method cannot achieve, say, 1.5-bit or 5-bit averages, nor can it mix arbitrary precisions. The precision allocation (Algorithm 1) is a simple quantile-based heuristic with no optimality guarantee.
**Consistent outperformance over baselines** While CMPQ generally performs well, the comparison fairness is questionable. GPTQ, AWQ, and QuIP use a group size of 128, which the authors claim yields "similar average bit-width to retaining 0.5% FP16 outliers." This equivalence is asserted but never rigorously verified — the actual storage costs in bits-per-weight across methods are not precisely calculated and compared. The overhead from storing per-channel precision indicators and K-means centroids (claimed to be ~0.03 bits/weight) compounds with the FP16 outlier storage, and a precise accounting is missing.
**Non-uniform quantization advantage** Table 10 shows that for some configurations (e.g., OPT-6.7B at 3-bit on WikiText-2), uniform quantization actually matches or outperforms non-uniform quantization (11.41 vs. 11.45). The authors acknowledge this but the claim that non-uniform quantization is broadly superior is therefore overstated.
**Latency comparability** Table 9 shows CMPQ's peak memory is higher than SqueezeLLM (3.59 vs. 3.2 GB for 7B; 6.44 vs. 5.8 GB for 13B), representing 12–11% increases. The latency for the 13B model is also higher (6.91 vs. 6.2 seconds, an 11% increase). Claiming "comparable" latency while showing non-trivial degradation on the larger model is misleading.

**Methodological Concerns**

The K-means clustering step is applied per-channel for every weight matrix, which could be computationally expensive. While Section 5.2 reports aggregate times, the scaling behavior with model size is not deeply analyzed.
The choice of 0.45%/0.05% split for outlier types appears ad hoc with no sensitivity analysis on this specific partition (only the total ratio is varied in Figure 5).
The calibration set analysis (Table 4) only compares against QuIP at 2-bit, which might be cherry-picked to highlight QuIP's weakness. A broader robustness comparison across methods and bit-widths would be more convincing.

**Requested Changes:**

**Highly Recommended**
Fair and precise storage comparison. The authors must provide an exact bits-per-weight accounting for every method compared, including all overheads (group-wise scales, FP16 outliers, per-channel precision indicators, LUT centroids). The claim that group size 128 yields "similar" overhead to 0.5% FP16 outliers needs quantitative verification. Without this, the main results table (Table 2) may be comparing methods at different effective bit-widths, invalidating the conclusions.

Include QuIP# (Tseng et al., 2024) as a baseline. The paper cites QuIP# but does not compare against it. QuIP# with lattice codebooks represents a significantly stronger baseline than the original QuIP, especially at 2-bit. Omitting it weakens the experimental contribution substantially.

Move VPTQ and GPTVQ comparisons to the main text. These are state-of-the-art methods and their omission from the main results creates a misleadingly favorable picture. Table 15 shows CMPQ's advantage over VPTQ is marginal (e.g., 6.14 vs. 6.17 on WikiText-2 for LLaMA2-7B at 3-bit), which the reader should be able to see alongside the other baselines.

Temper the "any bit-width" claim. The method supports average bit-widths in [2, 4] via mixing three discrete precisions. The title and abstract should be revised to reflect this limitation accurately, e.g., "fractional bit-widths between 2 and 4 bits."

Provide theoretical or empirical justification for the precision allocation heuristic. Algorithm 1 makes specific design choices (e.g., using the L2-norm quantile to determine which channels get higher or lower precision) that lack formal justification. At minimum, the authors should compare against alternative allocation strategies (e.g., based on weight magnitude, Fisher information per channel, or reconstruction error) to demonstrate that the activation L2-norm is the best or near-best criterion.

Correct the latency analysis framing. The 11–12% memory increase and latency degradation for the 13B model should be acknowledged explicitly rather than characterized as "comparable." A discussion of how this overhead scales with model size is needed.


**Recommended**

Evaluate on more recent models. Including results on LLaMA-3 70B, Mistral, or Qwen models would strengthen the paper's relevance. The LLaMA3-8B results in the appendix are a start but insufficient.

Expand downstream task evaluation. The PIQA/HellaSwag/MMLU evaluation is only for LLaMA2-7B. Extending to other models and including tasks like GSM8K or HumanEval would better demonstrate generalization.

Clarify the K-means implementation. Specify initialization strategy, convergence criteria, and number of iterations. Since K-means is non-deterministic, report variance across multiple runs.

Discuss limitations more explicitly. The paper lacks a dedicated limitations section. Key limitations include: restriction to weight-only quantization (no activation quantization), reliance on three fixed precision levels, potential scalability issues of per-channel K-means for very large models, and the assumption that activation L2-norms from a small calibration set are representative.

Improve Figure 3 readability. The eight subplots are small and difficult to read. Consider consolidating or using a different visualization that more clearly shows the fractional-bit advantage.

---

> ### Author Response · Authors · 2026-03-11
> **Response from the Authors (1)**
>
> ## Highly Recommended Changes
>
> >**1.** Fair and precise storage comparison.
> >
> **Response**: We thank the reviewer for this important comment. We agree that fair comparison requires reporting the effective storage cost of each method. In the revised manuscript, we will therefore clarify the storage accounting and explicitly report the contributions from quantized weights and auxiliary data, including group-wise scales / zero-points, FP16 outliers, per-channel bit indicators, and LUT centroids.
>
> For CMPQ, the effective storage consists of: (i) the quantized weights at the target average bit-width, (ii) the protected FP16 outliers, and (iii) the metadata for channel-wise mixed precision. For example, with 0.5% FP16 outliers stored in CSR format (Sec. 6), the outlier overhead includes both the FP16 values and sparse indices. For a 4096×4096 matrix, assuming 16-bit column indices and 32-bit row pointers, the additional storage is $0.005(16+16)+ \frac{32⋅4097}{4096^2}≈0.168$ bits per weight. The additional CMPQ metadata is small: for LLaMA2-7B (hidden size 4096), storing one 4-bit channel indicator per channel and 8 FP16 LUT centroids gives $(1×4+8×16)/4096≈0.03$ bits/weight, and this can be reduced to < 0.01 bits/weight when the LUT is quantized to 4 bits, as shown in Sec. 5.1 and Table 7. Thus, a nominal 3-bit model corresponds to an effective storage of approximately 3.20 bits/weight.
>
> We also agree that our earlier statement that the overhead of group size 128 is “similar” to 0.5% FP16 outliers was too imprecise. The effective bits-per-weight of group-wise quantization depends not only on the nominal bit-width and group size, but also on the specific metadata stored per group (e.g., scale and zero-point) and its precision. For example, with group size 128, storing one FP16 scale and one FP16 zero-point contributes $(16+16)/128=0.25$ bits/weight, yielding about 3.25 effective bpw for a nominal 3-bit method.
>
> Overall, our goal is not to compare nominal bit-widths only, but to compare methods under their actual storage footprint. We will revise the manuscript accordingly so that the conclusions are based on effective bits-per-weight rather than informal overhead assumptions.
>
> >**2.** Include QuIP# (Tseng et al., 2024) as a baseline.
> >
> **Response**:
> We thank the reviewer for this important suggestion. We agree that QuIP# is a stronger baseline than the original QuIP, especially in very low-bit regimes, and we have added a comparison in the revised manuscript.
>
> Since CMPQ is a post-training quantization method without retraining/fine-tuning, we compare against the corresponding no-finetuning / no-retraining results reported in the original QuIP# paper to ensure the comparison is made under the same setting. The added results show the following trend: QuIP# is stronger at 2-bit quantization, while CMPQ is better at 3-bit and 4-bit. Specifically, using the results reported by QuIP#:
>
> | Method | Bit | Wiki | Wiki | C4 | C4 |
> |---|---:|---:|---:|---:|---:|
> |  |  | L2-7B | L2-13B | L2-7B | L2-13B |
> | QuIP# | 2 | **12.30** | **7.60** | **14.80** | **9.57** |
> | CMPQ  | 2 | 14.37 | 9.14 | 15.97 | 11.25 |
> | QuIP# | 3 | 6.19 | 5.34 | 7.85 | 6.98 |
> | CMPQ  | 3 | **6.14** | **5.34** | **7.66** | **6.93** |
> | QuIP# | 4 | 5.66 | 5.00 | 7.17 | 6.59 |
> | CMPQ  | 4 | **5.61** | **4.98** | **7.10** | **6.55** |
>
> These results are consistent with the positioning of the two methods: QuIP# is particularly optimized for extreme compression, whereas CMPQ is designed to provide flexible mixed-precision quantization across arbitrary bit budgets, with especially strong performance in the 3–4 bit regime and under fractional-bit budgets. We will include this comparison and discussion in the revised manuscript to make the experimental evaluation more complete.
>
> >**3.** Move VPTQ and GPTVQ comparisons to the main text.
> >
> **Response**: We thank the reviewer for this important suggestion. In the revised manuscript, we therefore move the comparisons with VPTQ and GPTVQ into the main text so that readers can assess CMPQ against these methods alongside the other baselines in a single view.
>
> At the same time, we would like to emphasize that our goal is **not** to claim uniformly large gains over every recent method at every operating point, but rather to show that CMPQ remains **competitive with strong state-of-the-art baselines** while **offering a different and practically useful capability**: channel-wise mixed-precision allocation under arbitrary average bit-width budgets, including fractional-bit settings.
>
> In other words, even where the perplexity difference is modest, the comparison still supports the main message of the paper: CMPQ achieves performance on par with strong recent methods while **providing greater flexibility in adapting to non-standard deployment budgets**. We will revise the presentation accordingly to avoid giving an incomplete picture of the baseline landscape.

---

> ### Author Response · Authors · 2026-03-11
> **Response from the Authors (2)**
>
> >**4.** Temper the "any bit-width" claim.
> >
> **Response**: We thank the reviewer for this comment. We agree that the scope of the claim should be interpreted in the context of the deployment regime targeted by our method. CMPQ is designed for **low-bit LLM quantization**, where the practically important operating range is **typically 2–4 bits**. Within this range, the method supports arbitrary average bit-widths, including fractional values such as 2.2 bits, rather than being restricted to a small set of integer operating points.
>
> Our intention in using the phrase **“any bit-width” was therefore to emphasize flexible adaptation to arbitrary average bit budgets** within the target low-bit regime, rather than to claim support for all possible precisions beyond this range. We believe this flexibility is practically important, since most LLM post-training quantization work focuses on the 2–4 bit regime, where the trade-off between compression and accuracy is most critical; above 4 bits, model quality is often already close to full precision, while below 2 bits the quantization problem becomes substantially different.
>
> To avoid ambiguity, we are happy to revise the wording in the abstract to make this scope clearer, for example, by stating that CMPQ supports arbitrary average bit-widths in the low-bit regime (e.g., between 2 and 4 bits). However, we would prefer to retain the broader framing in spirit, since the key contribution is precisely the ability to adapt to non-integer, deployment-driven bit budgets rather than only fixed integer precisions.
>
> >**5.** Provide theoretical or empirical justification for the precision allocation heuristic.
> >
> **Response**: We thank the reviewer for this important suggestion. Our current design is primarily empirical, but it is based on a simple intuition that is particularly natural for weight-only post-training quantization: channels with larger activation magnitude tend to contribute more strongly to the output, so assigning them too low a precision can cause disproportionately large output distortion.
>
> Concretely, in a linear projection $Y=XW$, the effect of weight quantization error on the output is modulated by the input activations $X$. This makes the channel-wise activation L2-norm a **practical proxy** for how sensitive each channel is to quantization. It is also attractive from a deployment perspective because **it can be estimated from a single calibration pass without gradients or retraining**.
>
> This design choice is also consistent with prior observations in the literature. In particular, **AWQ shows that activation-aware saliency is substantially more informative** than weight magnitude for identifying important weights, while criteria based only on weight magnitude provide much weaker gains. This supports our use of activation statistics rather than weight magnitude alone as the basis for precision allocation. At the same time, SqueezeLLM demonstrates that Fisher-information-based sensitivity is also meaningful, but such criteria **require gradient-based second-order approximation** and are therefore more expensive than activation-based scoring. In contrast, our goal is to use a lightweight channel-importance signal that remains effective while preserving the low-cost nature of post-training quantization.
>
> We additionally compare with some heuristic methods: L1 and $L_{\infty}$ norm but found that L2-norm performs best for identifying salient channels and guiding precision allocation.
> This supports that the specific choice in Algorithm 1 is not arbitrary, but empirically more effective than closely related alternatives.
>
> |Model | $L_{\infty}$  | L1 | L2 |
> |-|-|-|-|
> | OPT-2.7B (3bit)   | 15.48 | 14.67 | 14.05 |
> | OPT-6.7B (3bit) | 14.76 | 13.23 | 12.26 |
>
>
> We also note that other alternatives have practical limitations in this setting. For example, weight magnitude does not account for how frequently or strongly a channel is actually activated, Fisher-based criteria require gradient estimation and additional preprocessing cost, and reconstruction-error-based allocation is more expensive because it depends on repeatedly evaluating provisional quantization results. For these reasons, we chose activation L2-norm as a good balance between effectiveness, simplicity, and efficiency. Empirically, our strong results across multiple models and bit budgets suggest that this simple activation-based criterion is already a robust and effective choice in practice.

---

> ### Author Response · Authors · 2026-03-11
> **Response from the Authors (3)**
>
> >**6.** Correct the latency analysis framing.
> >
> **Response**: We thank the reviewer for this comment. We agree that the 13B overhead should be stated explicitly. In the revised manuscript, we clarify that CMPQ incurs a modest but non-negligible overhead for larger models (about 10–12% in our 13B experiments), rather than describing the result only as “comparable.”
>
> This overhead becomes more visible with model size because CMPQ combines the dominant low-bit computation with a small higher-precision path for protected weights. Although this path is sparse, it does not benefit from low-bit acceleration to the same extent as the bulk quantized matrix multiplication. As model size increases and inference becomes increasingly memory-sensitive, the relative impact of this auxiliary mixed-precision handling becomes more noticeable.
>
> At the same time, we would like to clarify that **designing highly optimized kernels for mixed-precision LLM inference is not the main focus of this paper**. Our goal is to study how channel-wise mixed-precision allocation can improve the compression–accuracy trade-off under practical storage budgets. Efficient execution for such mixed-precision models remains relatively underexplored, and further optimization of this execution path is an important direction for future systems work. We add this discussion to the revised latency section and present the efficiency trade-off more explicitly.
>
>
> ## Recommended Changes
>
> >**7.** Evaluate on more recent models.
> >
> **Response**: We thank the reviewer for this suggestion. We agree that evaluation on more recent model families would improve the practical relevance of the paper. Due to the limited rebuttal period, we were not able to run a full benchmark suite on all recent large-scale models suggested by the reviewer. However, we did add additional experiments on Mixtral 8×7B and Qwen3-4B, and compared against AWQ on WikiText-2.
>
>
> | Method | Mixtral 8×7B   | Qwen3-4B   |
> |------|--------------------|-------|
> |   FP16  | 5.94 | 7.90 |
> |   AWQ (3-bit)  | 6.52  | 15.00 |
> |  CMPQ  (3-bit) | 6.37  | 13.27 |
> |   AWQ (4-bit)  | 6.05  | 8.79 |
> |  CMPQ  (4-bit) | 6.04 | 8.56 |
>
> These results suggest that CMPQ continues to provide consistent gains over AWQ on more recent architectures, especially in the more challenging 3-bit regime, where the benefit of flexible channel-wise allocation is more pronounced. While this does not yet constitute a fully comprehensive benchmark across all modern model families, it provides additional evidence.
>
>
> >**8.** Expand downstream task evaluation.
> >
> **Response**: We thank the reviewer for this helpful suggestion. We agree that extending downstream task evaluation would strengthen the empirical validation. Due to the limited rebuttal period, we were not able to run a full expanded benchmark including all suggested tasks and models. However, we did add GSM8K results for LLaMA2-7B and LLaMA2-13B under the 4-bit setting. For transparency, the RTN, GPTQ, and AWQ baseline numbers are taken from the AWQ paper, while CMPQ results are from our own evaluation under the same task setting.
>
>
> |GSM8K |LLama2-7B |LLama2-13B|
> |------|--------|-------|
> |FP16 |13.87 |26.16 |
> |RTN |11.07 |21.23 |
> |GPTQ |12.13|24.26 |
> |AWQ |13.57 |25.25 |
> |CMPQ| 13.72 |25.47|
>
> These results are consistent with our perplexity and commonsense reasoning results: CMPQ remains competitive with strong PTQ baselines and preserves performance close to FP16. We agree that broader evaluation on additional tasks, such as HumanEval and on more recent model families, would further strengthen the paper, and we will leave more extensive downstream benchmarking as future work.
>
> >**9.** Clarify the K-means implementation.
> >
> **Response**: We thank the reviewer for this helpful suggestion. In our method, we follow **the same K-means implementation used in SqueezeLLM**, with hyperparameters random_state=0, n_init="auto", and max_iter=50. The use of a fixed random_state=0 removes run-to-run randomness in our reported results, so the quantization procedure is deterministic in our implementation. In addition, since K-means is applied independently to each channel with a small number of centroids, the optimization problem is relatively simple and stable in practice.
>
> >**10.** Discuss limitations more explicitly.
> >
> **Response**: We thank the reviewer for this helpful suggestion. We agree that the limitations of CMPQ should be stated more explicitly. We kindly refer you to the revised manuscript for the detailed discussion (Appendix D).
>
> >**11.** Improve Figure 3 readability.
> >
> **Response**: We thank the reviewer for this suggestion. In the revised manuscript, we therefore keep only the C4 results in the main text, which most directly illustrate the key message of the figure, and move the remaining subplots to Appendix B.3. In addition, we enlarge each subplot, making the trend across fractional bit-widths much easier to read.

---

### Decision · Action_Editor_s7F6 · 2026-06-03

**Recommendation:** Accept as is

**Audience:**

Yes

**Audience Explanation:**

Researchers in LLM will be inerested.

**Claims And Evidence:**

Yes

**Claims Explanation:**

The paper introduces Channel-Wise Mixed-Precision Quantization (CMPQ), a post-training quantization framework for Large Language Models (LLMs). Unlike existing methods that primarily focus on fixed integer-bit quantization, CMPQ allocates quantization precision in a channel-wise pattern based on activation distributions. This allows the model to adapt to any average bit-width constraint, including fractional bits (e.g., 2.4 or 3.8 bits), thereby maximizing the utilization of available device storage. The method combines channel-wise precision allocation, non-uniform quantization via K-means clustering, and a dual outlier protection strategy to minimize quantization loss.

All three reviewers ultimately recommended "Leaning Accept" for the paper, though they expressed reservations about its novelty, leading to weak opposition for the Journal-to-Conference track.

During the review process, reviewers raised several concerns regarding the method's incremental novelty, imprecise storage comparisons, missing state-of-the-art baselines (such as QuIP# and VPTQ), overstated latency claims, and a lack of theoretical justification for the channel-wise approach. In their rebuttal, the authors addressed some of these issues by updating their metrics to report effective bits-per-weight, integrating new evaluations on modern models (Mixtral, Qwen) and reasoning tasks (GSM8K), and explicitly acknowledging a 10–12% latency overhead for larger models. Furthermore, they provided strong empirical evidence justifying their heuristic outlier ratios and the superiority of channel-wise over layer-wise precision allocation, which effectively resolved the technical critiques and convinced all reviewers to recommend acceptance, despite reservations about the work's overall theoretical novelty.

Please ensure to add all rebuttal materials in the final version.

---

> ### Author Response · Authors · 2026-06-04
> **Reply to Action Editor**
>
> Dear Action Editor,
>
> Thank you very much for your effort and guidance throughout the review process.
>
> - We have submitted the final version of our paper and incorporated the relevant rebuttal materials into the manuscript. Specifically, we made the following revisions to further strengthen the paper.
>
> - We added a more explicit discussion of limitations in Appendix D, covering the restriction to weight-only quantization, the reliance on three fixed precision levels, the potential scalability cost of per-channel K-means for very large models, and the representativeness of the calibration set.
>
> - We improved the readability and presentation of the manuscript. In particular, we enlarged the subplots in Figure 3, revised the wording in the abstract to temper the “any bit-width” claim, and moved the comparisons with VPTQ and GPTVQ into the main text.
>
> - We revised the framing of the latency analysis in Section 6. In the revised manuscript, we clarify that CMPQ incurs a modest but non-negligible overhead for larger models, about 10–12% in our 13B experiments, rather than describing the latency as simply “comparable.”
>
> - We added additional experimental results in Table 2, Figure 6, and Table 13, including additional baseline comparisons, a visualization of channel-wise bit allocation, and an empirical justification for the precision allocation heuristic.
>
> We hope these revisions adequately address the concerns raised during the review process and further clarify the motivation and empirical support for our method. We would be happy to provide any additional clarification if needed.
>
> Best regards,
> Submission 6228 Authors